# EVERYDAYMMQA: A MULTILINGUAL AND MULTIMODAL FRAMEWORK FOR CULTURALLY GROUNDED SPOKEN VISUAL QA

## ABSTRACT

Large-scale multimodal models achieve strong results on tasks like Visual Question Answering (VQA), but they often fail when queries require *culturally and visually grounded*, everyday knowledge, particularly in low-resource and underrepresented languages. To bridge this gap, we introduce Everyday Multimodal and Multilingual[1] QA (*EverydayMMQA*), a framework for creating large-scale, culturally-grounded datasets for spoken and visual question answering (SVQA). Using this framework, we developed *OASIS*, a multimodal dataset integrating speech, images, and text. With over $\sim$*0.92M images* and *14.8M QA pairs*, *OASIS* contains *3.7M spoken questions*, enabling four unique input combinations: speech-only, text-only, speech+image, and text+image. Focused on English and Arabic varieties, 18 countries, the dataset content is curated to reflect diverse, real-world situations. *OASIS* tests models on tasks beyond object recognition that involve pragmatic, commonsense, and culturally aware reasoning. We benchmarked four closed-source models, three open-source models, and one fine-tuned model. *EverydayMMQA* and *OASIS* together provide a framework, benchmark and training dataset for building multimodal LLMs for a comprehensive set of everyday tasks within cultural contexts. The *framework* and *dataset* will be made publicly available to the community.[2]

## 1 INTRODUCTION

Humans experience the world through multiple senses: sight, hearing, touch, smell, and taste. This multi-sensory integration is fundamental to how humans understand the surroundings and communicate. As large language models (LLMs) evolve, it is important to train them with multiple modalities: speech, text, and images, to mimic human interaction. For instance, when asking about an object, we often point to it while asking a question. In this scenario, we expect an AI assistant to process a multimodal triplet: the *visual information*, the *spoken information* (our question), and the contextual knowledge required to provide a *culturally appropriate response* (see Figure 1).

Crucially, this contextual knowledge is not universal: it is shaped by culture and language. Thus, successful multimodal reasoning requires grounding these signals in the specific cultural and linguistic context of the interaction, as gestures, phrasing, and interpretations vary dramatically across societies. However, current models are typically biased toward Western contexts (Nayak et al., 2024; Ananthram et al., 2025), often overlooking cultural and religious nuances in underrepresented languages. While recent resources (e.g.,

---

[1]For this study, we use the term *multilingual* to refer to English and Arabic varieties (Modern Standard Arabic (MSA), Egyptian and Levantine Arabic).

[2]anonymous.com

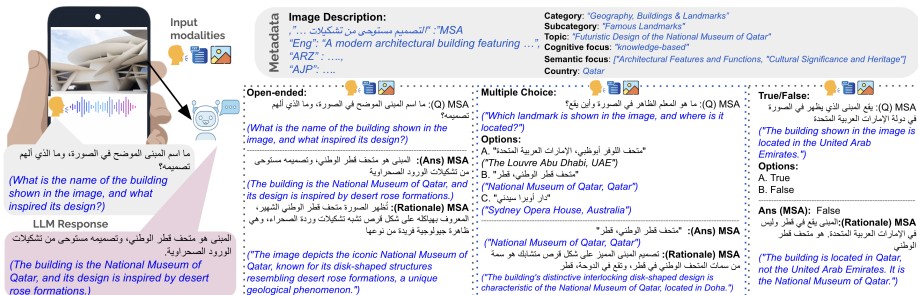

Figure 1: OASIS data sample, multimodal and multilingual QA around a culturally and visually grounded image. Example shows that the correct answer requires visual evidence from this image.

(Li et al., 2024a)) advance cultural evaluation, they generally focus only on text or image-text pairs, missing the essential *object–question–culture* triplet and rarely including spoken queries needed for real-world multimodal grounding.

Such lack of grounding has severe consequences for languages like Arabic. Cultural-awareness is vital given its dialectal diversity and country-specific uniqueness. Linguistic variation spans Modern Standard Arabic (MSA) and numerous regional dialects (e.g., Egyptian, Levantine), alongside differing traditions, religious expressions, and social norms across 22 Arab countries. An AI assistant ignoring the distinct traditions encountered by a tourist in Morocco versus Qatar risks producing irrelevant or even offensive outputs. Therefore, advancing equitable AI requires explicitly building multimodal, dialect-sensitive, and culturally grounded datasets.

To address these gaps, we first introduce *EverydayMMQA*, a flexible framework is language and location agnostic, that systematizes the creation of *scalable* multimodal and multilingual cultural resources. This framework enables efficient construction of datasets using its series of modules such as *(i)* culturally grounded *topic and query generation*, *(ii)* country-localized *image retrieval (iii) image filtering* and *metadata* generation, *(iv)* QA generation, *(v)* speech generation and recording, *(vi)* translation, and *(vii)* quality checking.

Using *EverydayMMQA*, we then developed *OASIS* — a large-scale, multimodal, and multilingual resource for training and evaluating cultural grounding and everyday reasoning. *OASIS* encompasses real-world QAs across 18 Arab countries, providing ∼**0.92M** images paired with **14.8M** question–answer pairs in English, MSA, and regional dialects. Each image includes **four QA types**: *one open-ended*, *one multiple-choice* (MCQ), and *two true/false* (T/F). Furthermore, the dataset comprises roughly **20K hours** of generated speech covering the entire corpus and **141 hours** of human recordings from a subset of the test splits. These support four input modalities: **text**, **speech**, **text+image**, and **speech+image**. Overall, *OASIS* provides a unique and comprehensive testbed for evaluating culturally diverse, everyday multimodal understanding, in addition to offering large-scale data splits for training.

Using *OASIS*'s comprehensive evaluation set, we benchmarked a suite of open and closed multimodal models (GPT-4.1, GPT-4o-audio, GPT-5, Gemini 2.5 Pro, Qwen2.5-3B,7B-Omni, and Phi-4) in a zero-shot setting. Different QA types probed specific capabilities: MCQ tested cultural knowledge, T/F measured hallucination, and open-ended judged real-time utility. Furthermore, we fine-tuned Qwen2.5-3B-Omni to inject cultural awareness using the speech, text, and image triplet modality.

Our **findings** shows visual grounding is the dominant lever, driving systematic performance gains across all models and languages. It narrows cross-lingual and dialectal disparities and acts as a modality equalizer, disproportionately benefiting speech and transcript inputs. Finally, with images and light fine-tuning, com-

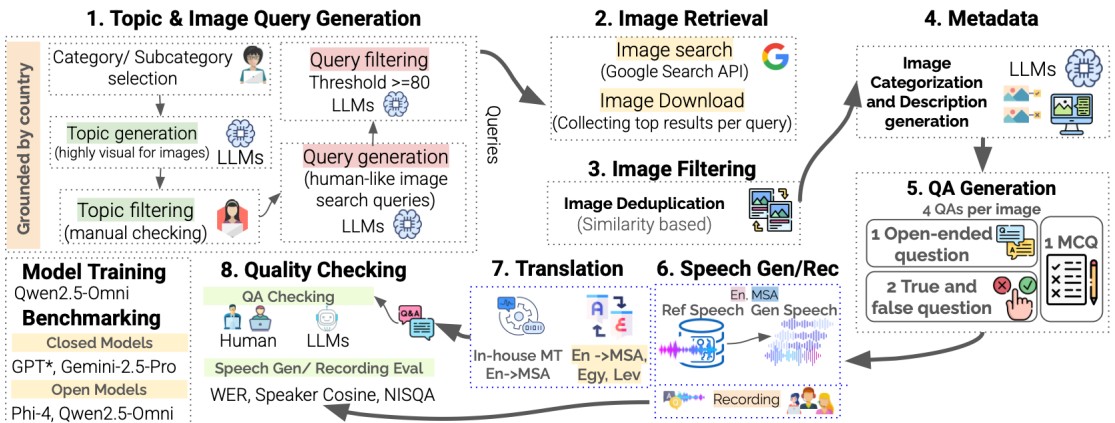

Figure 2: Proposed *EverydayMMQA* framework, *OASIS* dataset construction and experimental pipeline.

pact models can approach the performance of larger systems, highlighting the importance of cross-modal alignment and data quality for progress.

## 2 *EverydayMMQA* FRAMEWORK AND *OASIS* DEVELOPMENT

Existing AI resources lack the *object–question–culture* triplet and spoken queries, limiting real-world multimodal grounding for diverse languages like Arabic. This section details the *EverydayMMQA* framework and its use in creating *OASIS*, a multilingual, multimodal resource covering 18 Arabic-speaking countries to ensure cultural and dialectal diversity (Figure 2). In the subsections below, we provide details of each component of the *EverydayMMQA* framework. An example of a data point is also provided in Figure 1. More examples can be found in Appendix H. We provided all LLM prompts in Appendix C that are used in different modules of the framework.

### 2.1 CULTURALLY GROUNDED TOPIC & IMAGE QUERY GENERATION

We begin by designing a culturally grounded taxonomy, inspired by (Schwenk et al., 2022b; Vayani et al., 2025; Nayak et al., 2024), resulting in *9 categories* (defined as $\mathcal{G}$) and *31 subcategories* (Fig. 6). To curate a country-specific, culturally grounded image collection, we use web search and crafted user-oriented queries (Alam et al., 2025b) that reflect everyday information needs and natural variation, including typos and grammatical errors. To ensure strong alignment between the queries and the images, we followed two-step process: *topic generation and filtering*, followed by *query generation and filtering*.

**Topic Generation and Filtering:** For topic generation, we produce highly visual, country-wise grounded topics tied to each category and subcategory. We use LLMs to generate 10 topics per subcategory, yielding 310 topics per country. Formally, for each country $c$, let $\mathcal{C} = \{s_1, \ldots, s_{31}\}$ denote the set of 31 subcategories. For each $s_j \in \mathcal{C}$, an LLM generates topics $\mathcal{T}_{c,j} = \{t_{j,1}, \ldots, t_{j,10}\}$, and the full set of topics for country $c$ is $\mathcal{T}_c = \bigcup_{j=1}^{31} \mathcal{T}_{c,j}$ with $|\mathcal{T}_c| = 310$. Following, the topics are manually reviewed and revised by a human to remove (filter) generic or irrelevant cases and retain only those most suitable for image search.

**Query Generation and Filtering:** For generating large-scale naturalistic queries, we employed a set of LLMs ($\mathcal{L}$): GPT-4.1, Gemini-2.0-flash-001, and Claude-3.5-Sonnet, to maximize the diversity. The final query set for $(c, s_j, t_j)$ is then obtained by merging the outputs across all models, is $\mathcal{Q}_{c,j,t} = \bigcup_{l \in \mathcal{L}} \mathcal{Q}_{c,j,t}^{(l)}$, where each LLM ($l \in L$) generates $n_l$ distinct natural queries $\mathcal{Q}_{c,j,t}^{(l)} = \{q_1, q_2, \ldots, q_{n_l}\}$, for a country

$c$, and a subcategory $s_j$. Following generation, the queries are deduplicated, resulting in $\approx 6{,}100$ unique queries per country. Since image search for thousands of queries is computationally costly, we implemented a **relevancy filtering** step. In addition, we also wanted the queries to be location and culturally grounded. Hence, we prompt GPT-4o to assign a cultural-relevancy score to each country-based query $c$, subcategory $s_j$, and topic $t \in \mathcal{T}_{c,j}$. For each $q \in \mathcal{Q}_{c,j,t}$, the model returns a score $R_{\mathrm{LLM}}(c, s_j, t, q) \in [1, 100]$ reflecting the query's location and cultural fit in that context. We retain only queries with $R_{\mathrm{LLM}} \geq 80$, yielding the filtered set $\mathcal{Q}'_{c,j,t} = \{\, q \in \mathcal{Q}_{c,j,t} \mid R_{\mathrm{LLM}}(c, s_j, t, q) \geq 80 \,\}$. The threshold of $80$ was determined based on our manual inspection, where we aimed to balance coverage and specificity. In Table 3, we provide detailed statistics. On average across countries, the mean $\pm$ std is $176.1 \pm 13.5$. Across categories, it is $174.3 \pm 14.44$. The variance in the number of queries is a result of the image-search–based filtering process. This behavior is expected, some subcategories, such as *Famous Landmarks, Heritage & History*, are highly visual and associated with many distinct country-specific entities that yield diverse and relevant images. Hence, most candidate queries survive filtering. In contrast, some subcategories are less naturally grounded in images, for example, *eSports & Gaming* produced the fewest queries due to its lower visual relevance. This variation also reflects the underrepresentation of digital visual content for some countries. Finally, we aggregate across subcategories and topics to form the per-country pool $\mathcal{Q}'_c = \bigcup_j \bigcup_{t \in \mathcal{T}_{c,j}} \mathcal{Q}'_{c,j,t}$.

## 2.2 Country-Localized Image Retrieval

Using the filtered queries for each country (i.e., $\mathcal{Q}'_c$), we retrieved images via **country-localized search** (using Google Custom Search) with locale settings, safe search, and license options restricted to cc_publicdomain, cc_attribute, and cc_sharealike. For each query $q \in \mathcal{Q}'_c$, we collected the top $k$ results ($k \approx 20$–$40$), enforcing minimum-resolution and standard file type requirements. Formally, the retrieved image search set for country $c$ is $\mathcal{I}_c = \bigcup_{q \in \mathcal{Q}'_c} \mathcal{I}(q)$, where $\mathcal{I}(q)$ denotes the filtered set of image search results returned for query $q$. For each image, we stored provenance information (URL, query, country, category, subcategory, topic). In total, this phase retrieved $\sim$**4.3M** candidates from **18 Arab countries**, forming the global set $\mathcal{I} = \bigcup_c \mathcal{I}_c$.

## 2.3 Image Filtering

**Duplicate filtering.** We initially applied URL-based deduplication, and obtained $\sim$**2.4M** unique entries; however, due to timeouts and broken links, only $\sim$**1.4M** images were successfully downloaded. To further reduce redundancy, we applied exact and near-duplicate filtering. Specifically, features were extracted using a fine-tuned ResNet18 (He et al., 2016), and Euclidean distances between embeddings were used to identify duplicates ($|\mathbf{z}_i - \mathbf{z}_j|_2 \leq 1.5$). The final deduplicated set for country $c$ is $\mathcal{I}''_c \subseteq \mathcal{I}'_c$, and globally $\mathcal{I}'' = \bigcup_c \mathcal{I}''_c$. This yielded $\sim$**1.35M** unique images across all countries.

## 2.4 Metadata

**Image categorization and description generation.** For visual inspection, we developed a web portal and manually observed that the deduplicated set $\mathcal{I}''$ still contained non-representative items such as text-overlaid graphics, advertisements, charts, and screenshots. To filter these, each image was categorized using labels from set $\mathcal{Y} = \{$ *advertisement, photograph, illustration, other, chart/infographic, meme/text overlay, screenshot/ui capture, infographic*$\}$. The selection of these labels were motivated based on manual observation. In addition, we generated free-form text based image descriptions and assign suitability ($\mathcal{Z} = \{$*clarity, relevance, content*$\}$) labels. The description enrich the dataset and facilitate QA generation, while suitability labels indicate whether an image is appropriate for QA. We used LLM $l \in \mathcal{L}$, and formulated as a function $F_l : \mathcal{I}'' \to \text{text} \times \mathcal{Y} \times \mathcal{Z}$, where for each image $I \in \mathcal{I}''$, $F_l(I) = (d(I), y(I), z(I))$, with $d(I)$ denoting the free-text description, $y(I) \in \mathcal{Y}$ the categorization label, and $z(I) \in \mathcal{Z}$ the suitability label. For the final selection, we retained only images labeled as *photograph*, yielding $\mathcal{I}''' = \{I \in \mathcal{I}'' \mid y(I) =$

*photograph*}, totaling ∼**1.30M** images. Note that although suitability labels were generated, they were not used in the selection process and subsequent experiments. The detail prompt for $F_l$ is provided in Listing 6.

## 2.5 QA GENERATION

Developing robust multimodal and multilingual QA resources is challenging due to the significant time and cost involved in creating them manually (Changpinyo et al., 2023). To address this, recent progress in LLMs and VLMs offers a scalable means of automatically generating diverse QA pairs (Zhang et al., 2025). In this work, for each image $I$, we generate *four* questions, across three types: *(i)* 1 open-ended, *(ii)* 1 multiple-choice, and *(iii)* 2 true/false. **Open-ended** questions allow free-form answers, testing the model's ability to produce coherent, grounded responses despite the evaluation challenges posed by linguistic variability (Zhang et al., 2025). **MCQs** provide objective, reproducible evaluation, with plausible distractors designed to test fine-grained discrimination skills. **T/F** questions directly target hallucination and factual grounding. By including two per image, we measure both false positives (asserting details not present) and false negatives (missing present content) (Li et al., 2023).

We complement the three QA types with **semantic** ($\Sigma$) and **cognitive** ($\Xi$) profiles of questions. The 11 semantic categories (e.g., location, architecture, cultural heritage), derived by clustering annotator-written questions, align with prior VQA literature (Kafle & Kanan, 2017; Hudson & Manning, 2019). Each question is also labeled with a cognitive tag – *knowledge-based* or *commonsense-based* – aligning with benchmarks that distinguish external knowledge from everyday reasoning (Marino et al., 2019; Schwenk et al., 2022a; Zellers et al., 2019) (details in Appendix A.2).

We used an LLM $l \in \mathcal{L}$ to generate QAs conditioned on the category $g \in \mathcal{G}$, subcategory $s_j \in \mathcal{C}$, the image description $d(I)$, and the image $I$. Formally, we model QA generation as $G_l(g, s_j, d(I), I) = \{(p_r, a_r, r_r, \sigma_r, \xi_r)\}_{r=1}^4$, where $p_r \in \mathcal{P}$ is the $r$-th question, $a_r \in \mathcal{A}$ its answer, $r_r \in \mathcal{R}$ the rationale (reasoning) for the answer, $\sigma_r \in \Sigma$ the semantic label, and $\xi_r \in \Xi$ the cognitive label. During QA generation, some images triggered content filters and were excluded from the final dataset. For each image $I \in \mathcal{I}'''$, the per-image QA set is $\mathcal{S}(I) = \{(p_r, a_r, r_r, \sigma_r, \xi_r)\}_{r=1}^4$. At country-level $c$, QA set is $\mathcal{S}_c = \bigcup_{I \in \mathcal{I}_c'''} \mathcal{S}(I)$ and hence the global QA set is obtained by aggregating across all countries, i.e., $\mathcal{S} = \bigcup_c \mathcal{S}_c$, totalling **0.92M** images.

## 2.6 SPEECH GENERATION/RECORDING

The goal of the Spoken-QA task is to enable natural LLM interaction (speech-in → text-out) by leveraging acoustic cues. As large-scale natural spoken data is difficult to acquire, we utilize the XTTS-v2[3] (Casanova et al., 2024) model for speech synthesis in English and MSA. A high-quality benchmark set was manually recorded. Due to the lack of robust Arabic dialectal TTS models, our focus remains on English and MSA.

**Generation.** We created high-quality **reference voice resources** to enable speech generation. For English, we sampled 5–8 seconds segments from LibriTTS (Zen et al., 2019), selecting 10 utterances per speaker from 35 speakers, yielding 337 segments. For Arabic, we created a reference voice bank from QASR (Mubarak et al., 2021) and ADI17 (Shon et al., 2020), covering MSA and multiple dialects. Using the same 5–8 s criterion, we obtained 389 segments from 40 speakers in QASR and 69 manually reviewed segments from 7 unique ADI-17 speakers. For each question, we generated three audio samples in English and one in MSA, conditioned on randomly selected reference speakers. In future, we plan to extend this with more samples and additional dialects.

**Recording.** To complement synthetic speech, we collected a benchmark set of **human-recorded spoken QA** in English and MSA covering country-specific questions. Native Arabic and fluent English speakers recorded all questions types using our in-house platform under natural conditions, similar to (Alam et al.,

---

[3]https://huggingface.co/coqui/XTTS-v2

2025a). This effort resulted in ∼141 hours of recordings from 36 speakers (12 per language, an average duration of ∼5 seconds), providing a realistic reference for evaluating spoken QA.

## 2.7 TRANSLATION

We used an in-house LLM-based system to translate English into MSA. The performance of the system was evaluated on 11 datasets, achieving an average BLEU score of 25.11 compared to 19.62 with Google Translation system. For dialects, we compared GPT-4.1 direct translation with a two-step in-house pipeline (English → MSA → Dialect). As shown in Table 10 (Appendix A.6), BLEU scores and native speaker checks favored GPT-4.1, which we adopted for dialectal translation.

## 2.8 QUALITY CHECKING

**Manual Annotation.** We manually annotated a sample of QAs across all countries. QAs were rated for clarity on a five-point Likert scale, except True/False answers, which used a three-point scale. Annotators provided justifications for low scores. Rationales were also evaluated along two combined dimensions: (*i*) clarity & informativeness, and (*ii*) plausibility & faithfulness (Wang et al., 2023; Huang et al., 2023; Agarwal et al., 2024; Kmainasi et al., 2025). Annotation was conducted by native Arabic speakers using a dedicated interface, designed guidelines and expert supervision, with continuous quality checks. The detailed annotation guidelines are provided in Appendix G. In total, 13,728 samples from thirteen countries were annotated, yielding ∼110K annotations ($13,728 \times 4$ QAs $\times 2$ annotators).

**LLM-based Annotation.** We further employed Gemini-2.5-Pro and Llama-4-Scout-17B-16E-Instruct to replicate the human annotation tasks. Using the same guidelines and inputs (image $I$ and description $d(I)$), the models generated annotations for 34,930 samples (complete test split) across all 18 countries.

**Annotation Agreement** All QA evaluation metrics were rated on Likert scales, with the average of two annotators reported per item. To measure *inter-annotator* agreement on ordinal scales, we used the $r^*_{wg(j)}$ index (James et al., 1984). In Table 4 (Appendix A.3), we report both the human and LLM-based annotation agreement across open-ended, MCQ, and T/F types (LLMs evaluated on 18 and human on 13 countries). Our results show near-perfect agreement in answer consistency for T/F questions (LLMs: 0.97, humans: 0.99). MCQ scores are also very strong (LLMs: 0.87, humans: 0.95), while open-ended responses are slightly lower. The $r^*_{wg(j)}$ scores for answer quality in open-ended questions are 0.79 and 0.94 for LLMs and humans, respectively. Table 4 also reports the scores for question quality and rationales in both Likert scale and $r^*_{wg(j)}$ indices. These results confirm strong annotation consistency and high-quality QA pairs and rationales.

**LLM-Human Agreement.** We assessed the alignment between LLM annotations and human judgments across all QA formats. The correlations for question quality, answer quality, rationale plausibility & faithfulness, and rationale clarity & informativeness are 0.93, 0.87, 0.86, and 0.93, respectively, showing that LLMs can serve as reliable complementary annotators for quality assessment.

**Recording Quality:** We evaluated generated English and MSA audios using three standard metrics: *(i)* Word Error Rate (WER) for transcription accuracy, using Whisper-small (Radford et al., 2023); *(ii)* Speaker Cosine Similarity (SpkCos) for speaker consistency, based on embeddings from `spkrec-ecapa-voxceleb` (Ravanelli et al., 2021); and *(iii)* NISQA (Mittag et al., 2021), which predicts overall perceptual quality, including naturalness and distortion. For the human-recorded audio evaluation, we used Whisper-small for English and Fanar[4] for Arabic.

The quality evaluation (Appendix 5) reveals that generated English speech exhibits high perceptual quality (NISQA: 4.33), while Arabic is moderate (NISQA: 3.68), similar SpkCos indicates speaker consistency in both language. In contrast, for human recording, both English and Arabic audio show a relatively higher

---

[4]A publicly accessible ASR API: `https://fanar.qa/`.

WER. This is a critical as it reflects the natural complexities of real-world speech such as varying accents, background noise, among others that are not present in generated audio.

## 2.9 DATA ANALYSIS AND STATS

Figure 6 summarizes the key statistics of **OASIS**, which spans 18 MENA countries across text, speech, and image modalities. The dataset includes 3.7M QAs (open-ended, MCQ, T/F) in four language varieties, totaling ∼14.8M QA pairs. It exhibits balanced country coverage, with totals ranging from ∼36K (Qatar) to ∼64K (Morocco) (median ∼53K). Category representation is generally consistent (1–2.3K images per country), with notable cultural peaks (e.g., *Traditional & Regional Cuisines*) and lows (e.g., *Clothing & Fashion*). High-visibility categories (*Landmarks, National Symbols & Flags*) are consistently well-represented, ensuring balanced data with meaningful cultural variations. Detailed country and subcategory statistics are provided in Tables 8, and 9, in Appendix A.5.

**Audio.** The synthesized audio spans ∼20,279 hours in English and MSA, with average QA durations of 5s (English) and 6s (MSA), closely matching human recordings. Human recordings total ∼141 hours from 20 speakers across 9 countries. On average, each audio file corresponds to ∼52 tokens, considering ∼10 tokens per second of speech (Yeo et al., 2025; Guo et al., 2025).

**Images.** The image set has an average resolution of $1000 \times 1226$ px (width 372–2185, height 453–2415), confirming high visual quality. Tokens are computed tile-wise: each $512 \times 512$ tile contributes 85 base tokens plus 170 per tile, from which we derive both per-image and global totals.

**Data split.** Table 11 (in Appendix) reports the train, dev, test set splits for each country. Splits were created via subcategory-wise stratification, with ∼3.76% allocated to dev and test (about 2K samples per country, ∼34K total per split), and the remainder to training. A subset of the test set was human recorded and QAs manually checked.

## 3 EXPERIMENTS

**Models:** We evaluated six models from closed and open families: **GPT-4.1**, **GPT-5**, and **Gemini-2.5-Pro** (closed), and **Qwen-2.5-7B**, **Qwen-2.5-3B**, and **Phi-4** (open). This selection covers capabilities from frontier models to smaller, more accessible open-source ones.

**Benchmarking Setup:** All models are evaluated in a zero-shot setting under four input configurations: *T*, *S*, *T+I*, and *S+I*, with outputs always in text. We consider four task types per item: open-ended generation, MCQ, and two T/F variants (TF1, TF2). Since manual recordings cover only part of the test set, *S* and *S+I* evaluations are limited to items with open-ended and both T/F recordings (Table 7). We ran experiments based on each model's supported inputs, resulting in **108 distinct configurations** across models, modalities, and languages verities (See Table 12 in Appendix). Moreover, we fine-tuned the omni Qwen2.5-3b-omni with input combinations: S+I, T+I, T, and S. We also explored ASR transcripts ($T_r$) with speech. To ensure comparability, we fixed the prompt template and response schema for each task, kept decoding identical across models (temperature = 0, top-p = 1.0, fixed maximum output length).

**Training Setup:** Given the global **OASIS** dataset $\mathcal{S}$ and the training split $\mathcal{S}_{tr}$. We use $N_{tr} = 859.6$K base training images with $q_0 = 4$ questions per image; for language varieties $\mathcal{V} = \{\text{en}, \text{msa}, \text{arz}, \text{ajp}\}$ with input–modality counts $|\mathcal{M}_{en}| = |\mathcal{M}_{msa}| = 4$ and $|\mathcal{M}_{arz}| = |\mathcal{M}_{ajp}| = 2$, the number of training datapoints is $|\mathcal{S}_{tr}| = N_{tr} \times q_0 \times \sum_{v \in \mathcal{V}} |\mathcal{M}_v| = 859.6\text{K} \times 4 \times (4+4+2+2) \approx 41.26$M. Due to compute limits, we fine-tune on 6.67% of $\mathcal{S}_{tr}$ (≈ 2.75M datapoints), using Qwen2.5-3B with LoRA ($r=16$, $\alpha=32$), a learning rate of $2 \times 10^{-5}$, a maximum sequence length of 3072 tokens, and a single training epoch.

**Evaluation and Metrics.** We evaluate model performance on the **OASIS** test set using standard QA metrics. For semantic similarity, we report BERTScore F1 (Zhang et al., 2020). For open-ended questions, we use GPT-4.1 as LLM-as-judge following MT-Bench (Zheng et al., 2023), where responses are rated on a 1 to 10 rubric (helpfulness, relevance, accuracy, faithfulness). For MCQ and T/F questions, we compute accuracy.

Table 1: Evaluation results across languages and modalities. **F1** = F1 BERTScore, **Judge** = LLM-as-judge score, **Acc** = accuracy, **T** = text, **T+I** = text+image. Gemini = Gemini-2.5-pro. Underlined values denote the best text-only performance for each dialect, while bold values indicate the best text+image performance for Open-Ended Judge, MCQ, and True/False.

| Model | Modality | English | | | | | MSA | | | | |
|---|---|---|---|---|---|---|---|---|---|---|---|
| | | OE (F1) | OE (Judge) | MCQ (Acc) | TF1 (Acc) | TF2 (Acc) | OE (F1) | OE (Judge) | MCQ (Acc) | TF1 (Acc) | TF2 (Acc) |
| GPT-4.1 | T | 0.60 | 6.26 | 0.82 | 0.63 | 0.77 | 0.58 | 6.36 | 0.75 | 0.69 | 0.66 |
| | T+I | 0.73 | **8.60** | **0.98** | 0.97 | **0.99** | 0.62 | **8.36** | 0.96 | 0.96 | 0.98 |
| GPT-5 | T | 0.61 | 6.39 | 0.80 | 0.58 | 0.84 | 0.55 | 6.39 | 0.74 | 0.78 | 0.56 |
| | T+I | 0.66 | 8.46 | 0.98 | 0.97 | **0.99** | 0.57 | 8.10 | 0.97 | 0.97 | 0.98 |
| Gemini | T | 0.57 | 5.50 | 0.77 | 0.76 | 0.71 | 0.54 | 5.69 | 0.74 | 0.71 | 0.74 |
| | T+I | 0.63 | 7.14 | 0.97 | 0.93 | 0.98 | 0.56 | 6.90 | 0.96 | 0.94 | 0.98 |
| Qwen-7B | T | 0.57 | 5.11 | 0.70 | 0.61 | 0.64 | 0.53 | 4.45 | 0.56 | 0.77 | 0.41 |
| | T+I | 0.64 | 5.10 | 0.97 | **0.98** | 0.98 | 0.55 | 4.45 | 0.91 | 0.96 | 0.94 |
| Phi-4 | T | 0.55 | 5.01 | 0.66 | 0.43 | 0.78 | 0.51 | 3.71 | 0.50 | 0.61 | 0.54 |
| | T+I | 0.59 | 6.22 | 0.86 | 0.85 | 0.95 | 0.51 | 4.16 | 0.67 | 0.89 | 0.63 |
| Qwen-3B | T | 0.54 | 4.78 | 0.67 | 0.54 | 0.73 | 0.52 | 3.84 | 0.48 | 0.83 | 0.36 |
| | T+I | 0.50 | 5.27 | 0.35 | 0.97 | 0.97 | 0.52 | 4.91 | 0.39 | 0.96 | 0.87 |
| FT (Qwen-2.5-3B) | T | 0.73 | 6.39 | 0.92 | 0.91 | 0.88 | 0.64 | 5.85 | 0.89 | 0.90 | 0.86 |
| | T+I | 0.78 | 8.29 | **0.98** | **0.98** | **0.99** | 0.67 | 7.47 | **0.97** | **0.97** | **0.98** |
| | | Egyptian Arabic | | | | | Levantine Arabic | | | | |
| GPT-4.1 | T | 0.56 | 6.07 | 0.61 | 0.72 | 0.57 | 0.57 | 6.41 | 0.71 | 0.74 | 0.55 |
| | T+I | 0.62 | **8.30** | **0.81** | 0.95 | **0.98** | 0.62 | **8.39** | **0.92** | 0.96 | **0.98** |
| GPT-5 | T | 0.53 | 6.18 | 0.60 | 0.82 | 0.44 | 0.55 | 6.31 | 0.70 | 0.83 | 0.44 |
| | T+I | 0.55 | 7.86 | **0.81** | 0.96 | 0.98 | 0.57 | 8.03 | **0.92** | 0.97 | **0.98** |
| Gemini | T | 0.51 | 5.42 | 0.58 | 0.66 | 0.75 | 0.52 | 5.60 | 0.70 | 0.70 | 0.74 |
| | T+I | 0.52 | 6.62 | **0.81** | 0.93 | 0.97 | 0.55 | 6.85 | **0.92** | 0.94 | **0.98** |
| Qwen-7B | T | 0.48 | 4.07 | 0.45 | 0.75 | 0.45 | 0.49 | 4.23 | 0.55 | 0.80 | 0.40 |
| | T+I | 0.49 | 5.70 | 0.75 | 0.94 | 0.90 | 0.51 | 5.76 | 0.87 | 0.95 | 0.93 |
| Phi-4 | T | 0.46 | 2.94 | 0.39 | 0.66 | 0.47 | 0.47 | 3.24 | 0.45 | 0.70 | 0.48 |
| | T+I | 0.46 | 3.56 | 0.49 | 0.85 | 0.45 | 0.48 | 3.76 | 0.61 | 0.89 | 0.46 |
| Qwen-3B | T | 0.47 | 3.21 | 0.36 | 0.81 | 0.33 | 0.49 | 3.60 | 0.44 | 0.82 | 0.37 |
| | T+I | 0.45 | 4.18 | 0.27 | 0.95 | 0.75 | 0.46 | 4.32 | 0.32 | 0.96 | 0.81 |
| FT (Qwen-2.5-3B) | T | 0.64 | 5.85 | 0.72 | 0.89 | 0.84 | 0.64 | 5.96 | 0.84 | 0.90 | 0.85 |
| | T+I | 0.67 | 7.35 | 0.79 | **0.97** | 0.97 | 0.67 | 7.56 | 0.91 | **0.97** | **0.98** |

## 4 RESULTS

Table 1 comprehensively reports model performance across modalities, languages, and dialects (using BERTScore F1, LLM-as-judge, and accuracy). The results consistently show multimodal gains, near-ceiling accuracy on constrained tasks when images are present, and a narrowing of cross-lingual/dialect gaps. These strong initial findings motivate the subsequent analysis of modality impact and fine-tuning. We draw the following key observations:

**Image Modality: Shifting the Bottleneck** Providing the image yields large, consistent gains across all models. On constrained tasks (MCQ/TF), accuracies reach near ceiling (typically $\geq 0.93$). For open-ended answers, LLM-as-judge scores improve significantly (about $1\text{--}2+$ points), while BERTScore gains are modest. This indicates that visual evidence resolves recognition/grounding, shifting the primary bottleneck to faithful answer generation. Under multimodality, model rankings compress on simpler tasks, motivating the need for more complex, reasoning-dependent vision items to differentiate strong systems.

*Why does visual grounding help?* In text-only settings, MSA faces a significant linguistic challenge due to complex morphology, orthographic variability, and data sparsity, which amplify ambiguity (e.g., in referents and attributes). This makes text-only processing more challenging than in English. Based on our analysis, we observe that T+I performance for Arabic varieties is relatively higher than for English. To clarify this, we compute the absolute T→T+I differences across models (GPT-4.1 and GPT-5) for each language variety, as shown in Table 2. Across models, the T→T+I gain is consistently larger for Arabic (MSA, Egyptian, Levantine) than for English in accuracy-style tasks (e.g., MCQ and TF2), indicating that visual grounding helps in mitigating Arabic-specific linguistic challenges. Open-ended quality also increases, primarily via the judge metric, confirming better factual grounding. The small remaining gap between the English and MSA, likely stems from MSA generation issues (word choice, agreement) rather than failures in scene understanding.

Table 2: Absolute T→T+I gains across models and language variants.

| | GPT-4.1 | | | | GPT-5 | | | |
|---|---|---|---|---|---|---|---|---|
| Metric | English | MSA | Egy | Lev | English | MSA | Egy | Lev |
| MCQ (Acc) | +0.16 | +0.22 | +0.21 | +0.21 | +0.18 | +0.23 | +0.22 | +0.22 |
| TF2 (Acc) | +0.22 | +0.32 | +0.41 | +0.43 | +0.15 | +0.42 | +0.54 | +0.54 |

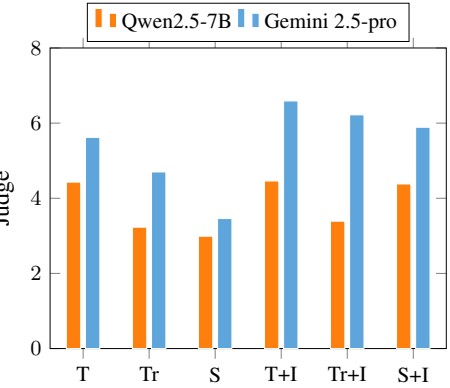 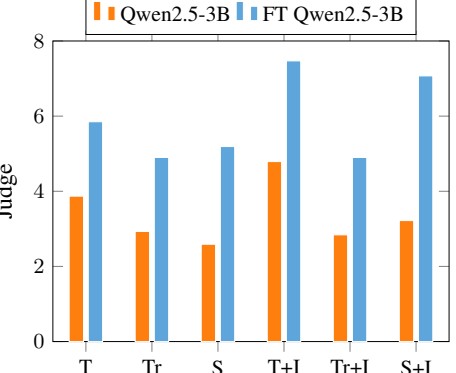

Figure 3: MSA Judge scores across modalities. Left: Qwen2.5-7B vs Gemini 2.5-pro. Right: Qwen2.5-3B vs its fine-tuned variant. English results are in the Appendix.

*Does grounding equalize dialectal difficulty?* Dialects suffer a disadvantage in text-only settings due to low-resourcedness and linguistic complexity, leading to lower scores than MSA. Visual grounding helps in reducing this "language complexity", anchoring the context and constraining answers. This results in increase in constrained task accuracy. As a result, the dialectal gap to MSA substantially narrows. While Levantine nearly converges with MSA, a slight residual MCQ gap for Egyptian suggests a dialect-specific generation issue rather than a failure of visual understanding.

**Closed *vs.* Open Models** Closed models perform significantly better than open models even in text-only settings, often achieving scores closer to the gold standard (e.g., GPT-4.1 *vs.* Qwen-2.5-7B on English and MSA MCQ). This suggests closed models leverage strong world knowledge and priors or benefit from broader pretraining and instruction tuning to make educated guesses and narrow the hypothesis space without visual evidence. This correctness without the image likely reflects priors and heuristics rather than genuine visual understanding, posing a deployment risk – models may answer confidently without grounding, increasing the chance of hallucination.

**Which metric best reflects post-grounding quality?** Results across all metrics consistently show that adding images helps. While MCQ and T/F tasks quickly saturate to near-ceiling accuracy, the LLM-as-judge score for open-ended answers increases substantially and retains headroom, making it the most informative signal for post-grounding quality and faithfulness. BERTScore, by contrast, improves only modestly and primarily reflects surface overlap. Practically, we use MCQ/TF as a basic check that the model is utilizing the image, and rely on LLM-as-judge to compare strong systems and analyze residual errors.

**Speech *vs.* Text Modalities:** Given the same query, **T** consistently outperforms **Tr** and **S** (See Figure 3). This is attributed to a two-stage noise stack in the speech data: *(i)* acoustic noise makes mapping raw **S** to the model's language space harder, and *(ii)* both ASR errors and normalization introduces noise (substitutions, formatting artifacts) in **Tr**, making it inferior to clean **T**. Details results in Appendix D, Table 13.

*Can visual grounding erase the input–channel penalty when the question is inherently visual?* Adding the image acts as a modality equalizer, recovering most of the performance loss from speech/transcripts and bringing them closer to original text. **I** supplies channel-agnostic evidence that anchors entities and

attributes, allowing the model to override acoustic and ASR noise. Consequently, the **S→S+I** input shows the largest gains, as the image reinstates critical cues and narrows the decoding search space.

**Fine-tuning and Findings:** Fine-tuning the Qwen2.5-3B model with the multimodal data transforms it into a stable multimodal responder (See Figure 3). This yields gains across all input channels, with the largest improvements seen in the raw speech and transcript. With images, **S+I** and **Tr+I**performance converges with **T+I** in English. Crucially, fine-tuning eliminates the fragility of **Tr+I** seen in the base model, indicating improved vision–text fusion and enhanced robustness to ASR artifacts. In summary, fine-tuning specifically stabilizes cross-modal alignment, making the small model competitive, especially with audio inputs.

Our findings confirm that image-centric questions are best answered when models "*see what the user sees*". Visual grounding drives large, systematic gains across all models and languages, effectively narrowing cross-lingual and dialect gaps. This process shifts residual errors from perception to faithful answer generation.

For evaluation, we recommend reporting $\Delta(\text{Text} \rightarrow \text{Image})$ alongside absolute metrics, and supplementing LLM-as-judge with calibration probes (ECE, Brier) to avoid over-reliance on saturated MCQ/TF scores. Future progress requires harder, visually confounded items (e.g., occlusions) and training that strengthens cross-modal alignment, particularly where the image acts as a strong equalizer for speech and transcripts.

## 5  RELATED WORK

**Omni – Large Multimodal Models.** Recent "omni" LMMs unify text, vision, audio, and video within a single architecture. Examples include QWEN2.5-OMNI (Xu et al., 2025), Phi-4 (Abdin et al., 2024), and closed models such as Gemini (Team et al., 2023). These systems handle diverse inputs and generate text or speech; For instance, QWEN2.5-OMNI employs a *Thinker–Talker* design with time-aligned multimodal encoding, achieving strong results on OmniBench (Li et al., 2024b). Microsoft's PHI-4-MULTIMODAL extends the Phi-4 recipe to vision–audio–text with multilingual reasoning, while earlier efforts such as KOSMOS-2 (Peng et al., 2023) foreshadowed this omni-modal direction. These models natively support our input regimes ($T$, $T+I$, $S$, $S+I$) with text outputs, but multilingual coverage, especially for Arabic, remains limited.

**Frameworks and Datasets.** Recent work has addressed culturally grounded multimodal resources, primarily through: *(i)* translating English corpora (PALO (Rasheed et al., 2025)), *(ii)* curating multilingual resources (Maya (Alam et al., 2024), Pangea (Yue et al.)), or *(iii)* adding speech to vision datasets (SPEECH-COCO). Examples include: Multilingual/Multimodal: PALO (translated image-text to 10 languages), Maya (8-language multimodal corpus), and Pangea (6M examples across 39 languages). Speech/ASR: SPEECH-COCO (∼600k spoken captions) and large ASR/ST resources like Common Voice and CoVoST 2. *Arabic-centric resources* are scarce, mostly image–text, such as CAMEL-Bench (∼29k VQA items) and Pearl (∼309k items for cultural understanding).

Our contribution differs by providing a unified framework for four modality setups (T, T+I, S, S+I), explicitly including dialect-aware Arabic alongside English, and offering a benchmark aligned for reproducible multi-model comparison (Table 15). We validate utility through baseline fine-tuning.

## 6  CONCLUSIONS AND FUTURE WORK

This paper presents the language-independent *EverydayMMQA* framework and the resulting *OASIS* dataset. OASIS is a large-scale, tri-modal resource covering 18 Arabic-speaking countries, comprising ≈0.92M images, 14.8M QAs, and 3.7M spoken QAs. We demonstrate its utility by benchmarking models and showing that a fine-tuned model consistently achieves higher accuracy on cultural knowledge, especially when questions are image-grounded (text+image, transcription+image, speech+image). Future work will focus on utilizing the full training set.

# 7 ETHICS STATEMENT

We do not foresee ethical concerns arising from this work. Images were collected in accordance with public-use licensing. For the manual recordings and annotations, contributors were compensated at standard hourly rates and were fully briefed on the tasks in advance.

# 8 REPRODUCIBILITY STATEMENT

We made every effort to ensure reproducibility. The main paper details the *EverydayMMQA* framework and the construction of *OASIS*, along with the training and evaluation setups. Appendices C.1, C.2, and C.3 provide the prompt instructions used to build *OASIS*, while Appendix A.1 outlines query preprocessing and ablation configurations. To further support replication, we include source code and scripts in the supplementary materials, and we will release all resources publicly[5].

# 9 LIMITATIONS

Due to computational resource constraints, we were unable to train models on the full dataset. With access to larger compute capacity, future work will leverage the complete dataset to fully explore its potential and further assess the benefits of large-scale multimodal training.

# 10 LLM USAGE

We primarily employed large language models as assistive tools for grammar and style refinement, and used GitHub Copilot for coding support.

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

# APPENDIX

# A  DETAILS OF THE *EverydayMMQA* AND *OASIS*

## A.1  TOPIC & IMAGE QUERY GENERATION

In Table 3, we provide detailed statistics for topic and query generation. We first generated 10 topics per subcategory using GPT-4o, resulting in 5,580 topics in total. After manually verifying country relevance, we retained 5,445 topics and removed 135 that were not aligned with the target countries. We provide the country name, category, subcategory and its all associated topics to generate queries (see Listings 2 and 3 for the prompts used). We asked each LLM to generate 50 queries per subcategory based on the provided topics. We then prompted GPT-4o to assess cultural relevance and assign a relevancy score from 1 to 100 for each query. We manually reviewed a sample of the queries and their scores to determine an optimal threshold for two purposes: *(i)* filtering out less relevant queries, and *(ii)* reducing the total number of queries. Finally, we selected only the queries with a relevancy score of $\geq 80/100$, yielding 97,678 queries in total. To understand per-subcategory query coverage, we analyzed the subcategory-wise distribution of queries across countries, as shown in Figures 4 and 5. From these figures, it is clear that some subcategories exhibit relatively higher coverage with low variance (e.g., Heritage & History), whereas others show lower coverage (e.g., Beverages).

Table 3: Statistics of the number of topics, filtered version after manual verification. Followed by query generation and filtering. Columns use ISO 2-letter country codes (DZ = Algeria, BH = Bahrain, EG = Egypt, IQ = Iraq, JO = Jordan, KW = Kuwait, LB = Lebanon, LY = Libya, MA = Morocco, OM = Oman, PS = Palestine, QA = Qatar, SA = Saudi Arabia, SD = Sudan, SY = Syria, TN = Tunisia, AE = United Arab Emirates, YE = Yemen). Rel. score: relevance score.

| Metric | DZ | BH | EG | IQ | JO | KW | LB | LY | MA | OM | PS | QA | SA | SD | SY | TN | AE | YE | Total |
|---|---|---|---|---|---|---|---|---|---|---|---|---|---|---|---|---|---|---|---|
| # Topics | 310 | 310 | 310 | 310 | 310 | 310 | 310 | 310 | 310 | 310 | 310 | 310 | 310 | 310 | 310 | 310 | 310 | 310 | **5,580** |
| Man. Verified | 304 | 305 | 310 | 303 | 297 | 301 | 306 | 307 | 310 | 309 | 294 | 310 | 287 | 310 | 283 | 303 | 310 | 296 | **5,445** |
| # Queries | 6,139 | 6,157 | 6,104 | 6,154 | 6,100 | 6,144 | 5,769 | 6,156 | 6,150 | 6,150 | 6,139 | 6,151 | 6,126 | 6,138 | 6,095 | 6,159 | 6,147 | 6,148 | **110,126** |
| Rel. $\geq 80$ | 5,405 | 5,347 | 5,531 | 5,465 | 5,483 | 5,356 | 5,167 | 5,519 | 5,509 | 5,654 | 5,387 | 4,875 | 5,459 | 5,521 | 5,476 | 5,448 | 5,521 | 5,555 | **97,678** |
| **Query Stats** | | | | | | | | | | | | | | | | | | | |
| Max | 192 | 195 | 196 | 191 | 194 | 195 | 196 | 193 | 197 | 196 | 191 | 200 | 196 | 195 | 195 | 193 | 197 | 197 | |
| Min | 139 | 140 | 152 | 146 | 146 | 120 | 107 | 150 | 147 | 161 | 146 | 75 | 140 | 153 | 144 | 138 | 125 | 140 | |
| Avg | 174.4 | 172.5 | 178.4 | 176.3 | 176.9 | 172.8 | 166.7 | 178.0 | 177.7 | 182.4 | 173.8 | 157.3 | 176.1 | 178.1 | 176.6 | 175.7 | 178.1 | 179.2 | |
| Std | 13.0 | 13.7 | 12.3 | 12.4 | 11.6 | 15.3 | 24.2 | 11.5 | 11.9 | 9.7 | 12.3 | 31.1 | 15.5 | 11.4 | 12.3 | 14.4 | 16.0 | 12.1 | |

## A.2  SEMANTIC AND COGNITIVE PROFILE FOR QA

We complement the three QA types with a **semantic** and **cognitive** profile of questions grounded in the image content. To obtain semantic type information, we selected two images per subcategory and manually wrote questions. For each image, annotators (several authors of the paper) independently inspected the content and drafted candidate questions. We then clustered the questions by semantic similarity across annotators and prompted an LLM to assign free-form labels. The resulting clusters were assigned one or more of the following labels: (*Location & place identification; Scene interpretation & context; Architectural features & functions; Cultural significance & heritage; Traditional clothing & attire; Tourism & cultural activities; Event & activity type; Objects, animals & food recognition; National symbols & identity; Visual attributes; Recreational activities & facilities*). This follows prior VQA practice of organizing questions by *semantic types* to enable targeted analysis (Kafle & Kanan, 2017; Hudson & Manning, 2019). In parallel, each question received a *cognitive focus* tag, either *knowledge-based* (requiring external/world knowledge) or *commonsense-based* (requiring everyday reasoning), which is in line with benchmarks that explicitly separate knowledge-intensive and commonsense reasoning (Marino et al., 2019; Schwenk et al., 2022a; Zellers

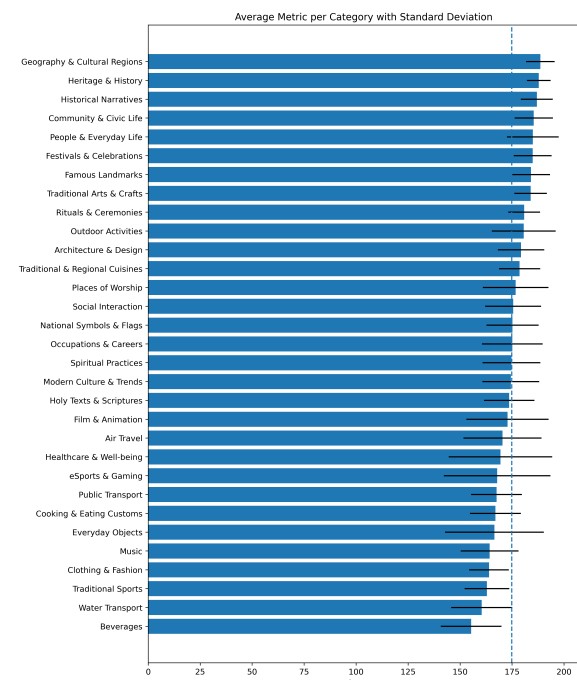

Figure 4: Average per category with standard deviation as error bars. Categories are sorted by average value. The dashed line indicates the global average across all categories.

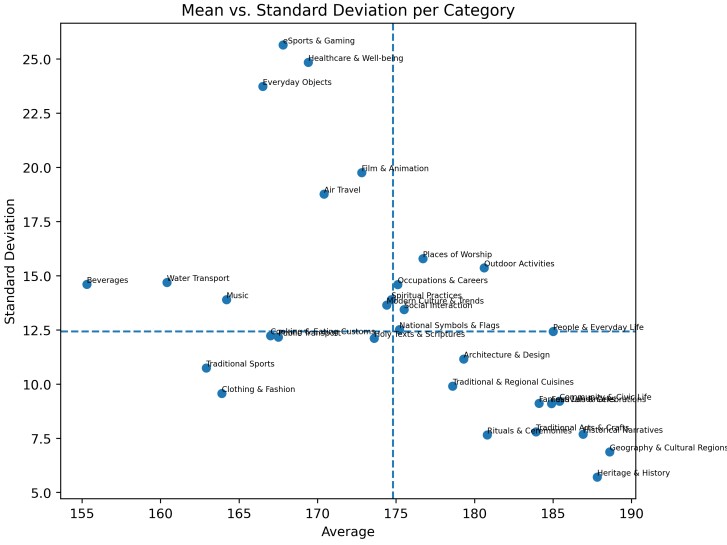

Figure 5: Scatter plot of mean score versus standard deviation for each category. The vertical dashed line indicates the global average score (174.8), and the horizontal dashed line indicates the median variability across categories, highlighting which categories are more visually grounded.

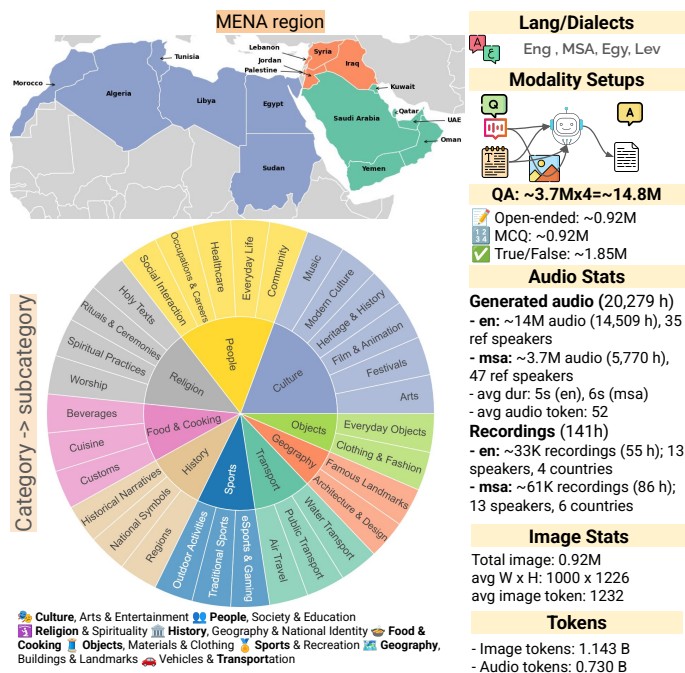

Figure 6: *OASIS* dataset overview: geographic coverage across 18 Arab countries, languages and dialects, modality setups (text, image, speech), QA types, audio durations, token counts, and per-(sub)category distributions.

et al., 2019). Our semantic labels also align with established vision domains widely used for scene/place, attributes, and landmarks (Zhou et al., 2018; Patterson & Hays, 2012; Weyand et al., 2020), facilitating transfer and comparison.

## A.3 ANNOTATION AGREEMENT FOR QA

We adopt the $r^*_{wg(j)}$ index (James et al., 1984) to measure agreement on ordinal Likert ratings. This compares observed variance in annotator ratings to the maximum possible variance under complete disagreement: $r^*_{wg(j)} = 1 - \frac{S_X^2}{\sigma^2_{\text{mv}}}$, where $S_X^2$ is the observed variance and maximum variance, $\sigma^2_{\text{mv}}$. For the 5-point Likert scale. $\sigma^2_{\text{mv}} = 0.5(X_U^2 + X_L^2) - [0.5(X_U + X_L)]^2$, with $X_U = 5$ and $X_L = 1$.

In Table 4, we report annotation agreement based on the Likert scale values for both human- and LLM-based annotations across three QA types such as open-ended, MCQ, and true/false (T/F) with LLMs evaluated across 18 countries and humans across 6 countries. Unless noted, quality means (A.Q., Q.Q., R.C.I., R.P.F.) are on a 1–5 scale; the T/F answer quality (A.Q.) uses a 1–3 scale. After linear rescaling, LLM T/F A.Q. of 2.90–2.95 (1–3) corresponds to approximately 4.80–4.90 on a 1–5 scale, and human T/F A.Q. of ∼2.99–3.00 maps to ∼4.99–5.00 indicating higher aggrement for binary answers. MCQ remains very strong across all mean quality dimensions (≈4.64–4.95 on 1–5), with open-ended close behind. rwg* score (0–1 scale) is uniformly high, ∼0.79–0.99 for LLMs and ∼0.95–0.99 for humans, with humans generally higher, especially on T/F. Overall, the high agreement scores suggest strong annotation consistency and high-quality QA pairs and rationales.

Table 4: Annotation agreement scores for LLM and human annotations. The values for answer and question qualities, such as A.Q. and Q.Q., range between 1-5 except A.Q. for true/false 1-3. The values $r^*_{wg}$ scores ranges between 0-1. *A.Q.* = Answer Quality Mean, *A.Q.* $r^*_{wg}$ = Answer Quality Inter-rater Agreement ($r^*_{wg}$), *Q.Q.* = Question Quality Mean, *Q.Q.* $r^*_{wg}$ = Question Quality Inter-rater Agreement ($r^*_{wg}$), *R.C.I.* = Rationale Clarity & Informativeness Mean, *R.C.I.* $r^*_{wg}$ = Rationale Clarity & Informativeness Inter-rater Agreement ($r^*_{wg}$), *R.P.F.* = Rationale Plausibility & Faithfulness Mean, *R.P.F.* $r^*_{wg}$ = rationale plausibility & faithfulness inter-rater agreement ($r^*_{wg}$).

| QA Type | A.Q. | A.Q. $r^*_{wg}$ | Q.Q. | Q.Q. $r^*_{wg}$ | R.C.I. | R.C.I. $r^*_{wg}$ | R.P.F. | R.P.F. $r^*_{wg}$ |
|---|---|---|---|---|---|---|---|---|
| **LLM-based Annotation (18 countries)** | | | | | | | | |
| Open-ended | 4.680 | 0.788 | 4.916 | 0.958 | 4.938 | 0.979 | 4.756 | 0.834 |
| MCQ | 4.823 | 0.868 | 4.926 | 0.963 | 4.949 | 0.979 | 4.839 | 0.880 |
| T/F-0 | 2.898 | 0.963 | 4.947 | 0.973 | 4.967 | 0.986 | 4.874 | 0.902 |
| T/F-1 | 2.945 | 0.980 | 4.969 | 0.984 | 4.982 | 0.993 | 4.922 | 0.940 |
| **Human Annotation (13 countries)** | | | | | | | | |
| Open-ended | 4.629 | 0.938 | 4.684 | 0.946 | 4.545 | 0.936 | 4.575 | 0.939 |
| MCQ | 4.657 | 0.946 | 4.700 | 0.952 | 4.591 | 0.947 | 4.600 | 0.944 |
| T/F-0 | 2.989 | 0.997 | 4.644 | 0.947 | 4.607 | 0.952 | 4.595 | 0.948 |
| T/F-1 | 2.993 | 0.997 | 4.147 | 0.916 | 4.602 | 0.948 | 4.577 | 0.945 |

## A.4 QA AUDIO

**Audio Question Evaluation.** We assessed audio quality using four standard metrics: *(i)* Word Error Rate (WER) for transcription accuracy; *(ii)* Speaker Cosine Similarity (SpkCos) for speaker consistency, based on embeddings from `spkrec-ecapa-voxceleb` (Ravanelli et al., 2021); and *(iii)* NISQA (Mittag et al., 2021), which predicts overall perceptual quality, including naturalness and distortion. For the transcription of the generated audio, both English and MSA, we used Whisper-small Radford et al. (2023). For the human-recorded audio transcription evaluation, we used Whisper-small for English and Fanar[6] for Arabic. Table 5 show the quality difference between the generated and human recorded audio.

Table 5: Objective evaluation of generated (Gen.) and human-recorded (Human) audio for English and MSA. **WER** = Word Error Rate (lower is better), **SpkCos** = Speaker Cosine Similarity (higher is better), **NISQA** = Non-Intrusive Speech Quality Assessment (higher is better).

| Language | WER (Gen.) ↓ | SpkCos (Gen.) ↑ | NISQA (Gen.) ↑ | WER (Human) ↓ |
|---|---|---|---|---|
| English | 6.19 | 0.58 | 4.33 | 22.25 |
| MSA | 9.85 | 0.57 | 3.68 | 22.65 |

## A.5 DATA STATISTICS

Figure 7 shows the distribution between commonsense and knowledge-based questions. Overall, the dataset is balanced (51.9% knowledge-based vs. 48.1% commonsense). Open-ended questions are mostly knowledge-based (72.9%), while true/false questions are mostly commonsense (57.5%). Multiple-choice questions are almost evenly split, with 49.6% being knowledge-based and 50.4% being commonsense-based.

Table 6 reports the mean and standard deviation of of text length (in words) across languages. In English, the statistics are consistently the highest across all cases, such as, descriptions (32.11±8.85), description reasons (37.84±5.59), questions (12.43±2.91), and rationales (19.56±4.28). Within the Arabic varieties, lengths decrease roughly as MSA > Egyptian ≈ Levantine (e.g., description lengths: 26.66±7.63, 24.75±9.71,

---

[6]A publicly accessible ASR API: `https://fanar.qa/`.

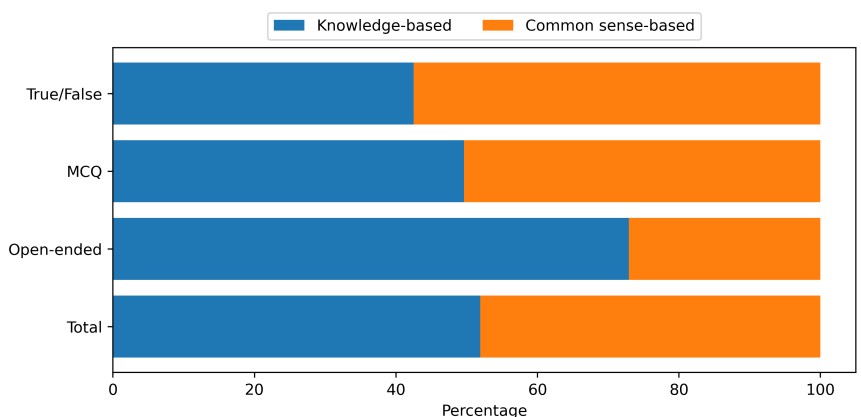

Figure 7: Distribution of commonsense and knowledge based for the whole dataset.

24.45±11.75). Answers are short across languages (EN 7.71, MSA 6.79, Egy 6.89, Lev 6.53 words on average) yet show large variance (std ≈ 9–11), indicating a mix of one-word and phrase-level responses. Levantine exhibits the greatest dispersion (e.g., question std 6.79; rationale std 8.58), suggesting greater stylistic heterogeneity. Overall, the dataset provides substantive rationales (∼16–20 words) and comparable question lengths across Arabic dialects, with English being more verbose.

In Tables 8 and 9, we report the subcategory-wise data distribution across countries. Overall, the number of images per country ranges from 40K to 60K, although for a few countries it is lower, such as Qatar with 36K images.

Table 6: Statistics of the image description, question, and rationals for language varieties. Numbers are represented as (mean±std).

| Lang | Description | Description Reason | Question | Answer | Rationale |
|------|-------------|--------------------|----------|--------|-----------|
| EN | $32.11 \pm 8.85$ | $37.84 \pm 5.59$ | $12.43 \pm 2.91$ | $7.71 \pm 11.01$ | $19.56 \pm 4.28$ |
| MSA | $26.66 \pm 7.63$ | $31.06 \pm 5.42$ | $9.95 \pm 2.88$ | $6.79 \pm 9.68$ | $16.90 \pm 4.18$ |
| Egy | $24.75 \pm 9.71$ | $28.73 \pm 5.64$ | $9.68 \pm 3.98$ | $6.89 \pm 9.58$ | $16.51 \pm 5.55$ |
| Lev | $24.45 \pm 11.75$ | $28.62 \pm 7.22$ | $8.85 \pm 6.79$ | $6.53 \pm 9.44$ | $15.57 \pm 8.58$ |

**Recordings statistics** In Table 7, we report the number of MSA and English recording samples per country. Overall, the evaluation set supports cross-country, cross-variety analysis despite uneven per-country and per-language distributions.

Table 7: Number of MSA and English recording samples associated with open-ended and T/F questions per country used in evaluation. Country codes: DZ = Algeria, BH = Bahrain, EG = Egypt, IQ = Iraq, JO = Jordan, KW = Kuwait, LB = Lebanon, LY = Libya, MA = Morocco, OM = Oman, PS = Palestine, QA = Qatar, SA = Saudi Arabia, SD = Sudan, SY = Syria, TN = Tunisia, AE = United Arab Emirates, YE = Yemen.

| | DZ | BH | EG | IQ | JO | KW | LB | LY | MA | OM | PS | QA | SA | SD | SY | TN | AE | YE |
|---|---|---|---|---|---|---|---|---|---|---|---|---|---|---|---|---|---|---|
| **# of MSA** | 654 | 1,200 | 1,739 | 1,571 | 1,990 | 1,138 | 1,996 | 1,997 | 1,990 | 1,991 | 1,986 | 1,995 | 1,995 | 2,001 | 2,004 | 1,324 | 1,997 | 2,000 |
| **# of English** | 1,153 | 1,329 | 2,014 | 1,143 | 1,113 | 1,355 | 1,375 | 1,387 | 1,108 | 1,100 | 702 | 1,995 | 1,680 | 2,001 | 1,016 | 1,249 | 1,016 | 1,218 |
| **Total** | 1,807 | 2,529 | 3,753 | 2,714 | 3,103 | 2,493 | 3,371 | 3,384 | 3,098 | 3,091 | 2,688 | 3,990 | 3,675 | 4,002 | 3,020 | 2,573 | 3,013 | 3,217 |

Table 8: Per-country counts (first half of categories). Abbreviations: Air = Air Travel; Arch = Architecture & Design; Bev = Beverages; Cloth = Clothing & Fashion; Civic = Community & Civic Life; Cook = Cooking & Eating Customs; Obj = Everyday Objects; Landm = Famous Landmarks; Fest = Festivals & Celebrations; Film = Film & Animation; Geo = Geography & Cultural Regions; Health = Healthcare & Well-being; Herit = Heritage & History; HistN = Historical Narratives; Script = Holy Texts & Scriptures; Modern = Modern Culture & Trends.

| Country | Air | Arch | Bev | Cloth | Civic | Cook | Obj | Landm | Fest | Film | Geo | Health | Herit | HistN | Script | Modern |
|---|---|---|---|---|---|---|---|---|---|---|---|---|---|---|---|---|
| algeria | 2,135 | 1,346 | 1,596 | 2,000 | 1,815 | 1,624 | 2,632 | 1,447 | 1,352 | 1,035 | 1,741 | 1,649 | 1,288 | 1,402 | 1,065 | 1,511 |
| bahrain | 1,778 | 1,429 | 1,231 | 2,113 | 1,452 | 1,371 | 2,059 | 1,199 | 1,245 | 1,198 | 1,528 | 1,543 | 1,159 | 1,180 | 959 | 1,033 |
| egypt | 2,384 | 1,923 | 1,739 | 2,141 | 2,025 | 1,504 | 2,666 | 1,843 | 1,589 | 1,444 | 2,596 | 1,877 | 1,546 | 1,844 | 1,126 | 1,274 |
| iraq | 1,925 | 1,257 | 1,536 | 2,317 | 2,018 | 1,365 | 2,278 | 1,151 | 1,339 | 970 | 1,966 | 1,715 | 1,294 | 1,359 | 930 | 1,517 |
| jordan | 1,864 | 1,779 | 1,899 | 2,382 | 1,664 | 1,636 | 2,400 | 1,634 | 1,612 | 1,438 | 2,367 | 1,357 | 1,949 | 1,918 | 1,290 | 1,489 |
| kuwait | 1,700 | 1,757 | 1,173 | 2,014 | 1,363 | 1,424 | 1,961 | 1,734 | 1,107 | 895 | 1,381 | 1,707 | 1,042 | 1,092 | 879 | 1,404 |
| lebanon | 1,941 | 1,937 | 1,780 | 1,963 | 1,952 | 1,756 | 2,094 | 1,644 | 1,315 | 1,290 | 2,183 | 1,803 | 1,730 | 1,702 | 1,405 | 1,887 |
| libya | 2,078 | 1,283 | 1,662 | 2,417 | 1,622 | 1,429 | 2,678 | 1,330 | 1,201 | 1,051 | 1,736 | 1,691 | 1,067 | 1,297 | 971 | 1,518 |
| morocco | 2,248 | 1,984 | 2,130 | 2,750 | 1,850 | 2,051 | 3,010 | 2,208 | 1,714 | 1,661 | 2,700 | 1,863 | 2,102 | 2,170 | 1,354 | 1,826 |
| oman | 2,064 | 2,391 | 1,418 | 1,853 | 2,092 | 1,427 | 1,963 | 1,826 | 1,342 | 1,381 | 2,169 | 1,628 | 2,080 | 1,818 | 1,260 | 1,638 |
| palestine | 2,164 | 2,028 | 1,655 | 1,716 | 1,247 | 1,820 | 2,150 | 1,682 | 1,433 | 959 | 2,725 | 1,876 | 1,597 | 1,930 | 1,456 | 1,483 |
| qatar | 1,556 | 1,948 | 886 | 1,581 | 981 | 747 | 2,111 | 1,331 | 670 | 1,546 | 860 | 1,149 | 1,065 | 942 |  | 1,202 |
| saudi_arabia | 2,189 | 1,536 | 1,705 | 2,504 | 1,834 | 1,889 | 2,744 | 1,872 | 1,669 | 1,380 | 2,088 | 1,360 | 1,726 | 1,675 | 1,342 | 1,863 |
| sudan | 1,811 | 1,576 | 1,558 | 1,980 | 1,651 | 1,570 | 2,598 | 1,538 | 1,417 | 874 | 1,519 | 1,595 | 1,418 | 1,405 | 1,141 | 1,249 |
| syria | 1,749 | 1,407 | 1,755 | 1,995 | 1,446 | 1,670 | 2,246 | 1,460 | 1,583 | 1,268 | 1,565 | 1,730 | 1,369 | 1,536 | 1,108 | 1,352 |
| tunisia | 1,426 | 1,831 | 2,376 | 2,418 | 1,615 | 1,907 | 2,079 | 1,738 | 1,725 | 1,267 | 1,768 | 2,167 | 1,683 | 1,601 | 1,117 | 1,410 |
| uae | 2,453 | 2,701 | 1,646 | 2,277 | 1,783 | 1,734 | 2,135 | 2,361 | 1,624 | 1,332 | 2,783 | 1,706 | 1,488 | 1,961 | 1,051 | 1,972 |
| yemen | 2,142 | 1,331 | 1,389 | 2,095 | 1,857 | 1,411 | 2,269 | 1,526 | 1,444 | 1,335 | 1,660 | 1,766 | 1,421 | 1,615 | 932 | 1,818 |

Table 9: Per-country counts (second half of categories). Abbreviations: Music = Music; Symbols = National Symbols & Flags; Jobs = Occupations & Careers; Outdoor = Outdoor Activities; People = People & Everyday Life; Worship = Places of Worship; Transit = Public Transport; Rituals = Rituals & Ceremonies; Social = Social Interaction; Spiritual = Spiritual Practices; Cuisine = Traditional & Regional Cuisines; Crafts = Traditional Arts & Crafts; Sports = Traditional Sports; Water = Water Transport; eSports = eSports & Gaming.

| Country | Music | Symbols | Jobs | Outdoor | People | Worship | Transit | Rituals | Social | Spiritual | Cuisine | Crafts | Sports | Water | eSports | Total |
|---|---|---|---|---|---|---|---|---|---|---|---|---|---|---|---|---|
| algeria | 1,536 | 955 | 2,057 | 1,769 | 1,936 | 1,478 | 2,063 | 1,435 | 1,675 | 1,303 | 1,644 | 1,814 | 1,565 | 2,258 | 1,252 | 50,378 |
| bahrain | 1,594 | 1,359 | 1,392 | 1,407 | 1,397 | 1,490 | 1,545 | 1,245 | 1,158 | 1,097 | 1,296 | 1,977 | 1,592 | 1,812 | 1,455 | 44,293 |
| egypt | 1,602 | 1,415 | 1,834 | 1,812 | 1,984 | 2,043 | 2,416 | 1,272 | 1,895 | 1,216 | 1,891 | 1,639 | 1,718 | 2,250 | 1,386 | 55,894 |
| iraq | 1,481 | 1,086 | 1,673 | 1,564 | 1,699 | 1,462 | 1,854 | 1,476 | 1,753 | 1,390 | 1,443 | 1,625 | 1,781 | 1,643 | 1,610 | 48,477 |
| jordan | 1,695 | 1,782 | 1,579 | 1,648 | 2,002 | 1,424 | 1,965 | 1,708 | 1,366 | 1,575 | 1,844 | 2,203 | 2,023 | 1,659 |  | 54,536 |
| kuwait | 1,706 | 1,417 | 1,329 | 1,982 | 1,527 | 1,311 | 1,469 | 912 | 1,233 | 928 | 1,207 | 1,686 | 1,533 | 1,780 | 1,339 | 43,992 |
| lebanon | 1,586 | 1,431 | 1,712 | 1,170 | 2,022 | 1,657 | 2,329 | 1,606 | 1,578 | 1,830 | 1,855 | 1,760 | 2,004 | 1,592 | 1,325 | 53,839 |
| libya | 1,787 | 1,076 | 1,687 | 1,722 | 1,766 | 943 | 2,104 | 1,385 | 1,463 | 1,091 | 1,534 | 1,677 | 1,835 | 1,917 | 1,437 | 48,455 |
| morocco | 1,867 | 1,862 | 1,826 | 1,785 | 1,960 | 1,643 | 2,526 | 1,603 | 2,020 | 1,501 | 2,266 | 3,039 | 2,194 | 2,338 | 1,677 | 63,728 |
| oman | 1,795 | 1,410 | 1,709 | 1,971 | 1,652 | 1,816 | 2,123 | 1,495 | 1,527 | 1,326 | 1,548 | 1,813 | 1,752 | 1,724 | 1,793 | 53,804 |
| palestine | 1,877 | 1,560 | 1,922 | 1,941 | 1,629 | 1,851 | 2,285 | 1,643 | 1,373 | 1,785 | 1,557 | 1,445 | 2,339 | 2,279 | 1,548 | 54,955 |
| qatar | 854 | 1,524 | 897 | 1,129 | 929 | 1,555 | 1,224 | 945 | 904 | 918 | 786 | 1,237 | 995 | 1,395 | 826 | 36,071 |
| saudi_arabia | 1,687 | 1,392 | 1,912 | 2,063 | 2,188 | 1,931 | 1,548 | 1,842 | 1,664 | 1,870 | 1,698 | 1,781 | 2,168 | 1,759 | 1,427 | 56,306 |
| sudan | 1,598 | 959 | 1,761 | 1,981 | 1,865 | 1,612 | 2,157 | 1,043 | 1,639 | 1,330 | 1,449 | 1,722 | 1,953 | 1,970 | 1,623 | 49,562 |
| syria | 1,687 | 1,077 | 1,420 | 1,889 | 1,796 | 1,347 | 2,004 | 1,293 | 1,738 | 1,396 | 1,327 | 1,501 | 2,037 | 2,227 | 1,583 | 49,561 |
| tunisia | 1,702 | 1,445 | 1,817 | 1,844 | 1,918 | 1,841 | 1,819 | 1,519 | 1,908 | 1,519 | 1,838 | 2,020 | 2,340 | 1,783 | 1,468 | 54,909 |
| uae | 1,792 | 1,684 | 1,951 | 2,082 | 2,087 | 1,830 | 2,007 | 1,634 | 1,780 | 1,853 | 1,604 | 2,018 | 2,084 | 2,118 | 1,517 | 59,048 |
| yemen | 1,964 | 1,234 | 1,915 | 1,690 | 1,550 | 1,345 | 2,335 | 1,542 | 1,676 | 1,238 | 1,620 | 1,756 | 2,236 | 1,887 | 1,678 | 51,677 |

## A.6 MACHINE TRANSLATION SCORES

In Table 10, we compare BLEU for direct English→Dialect (EN→DIA) *vs.* a two-step English→MSA→Dialect (EN→MSA→DIA) pipeline. For Levantine, the direct approach consistently outperforms the two-step pipeline across all test sets (avg. 13.81 vs. 7.74 BLEU). For Egyptian, results are mixed: the pipeline excels on *arzen* and *madar test nil 0 eg* (22.86 and 31.33 BLEU, respectively), while the

direct model leads on *madar test nil 1/2 eg*. These trends suggest that intermediate MSA can help for certain Egyptian benchmarks, whereas direct transfer is more reliable for Levantine.

Table 10: Comparison of GPT-4.1 performance (BLEU score) with the 2-step pipeline performance. The EN->Dia columns correspond to GPT-4.1 performance while the EN->MSA->Dia column correspond to the 2-step pipeline performance

| | Levantine | | | Egyptian | |
|---|---|---|---|---|---|
| Test Set | EN->MSA-> DIA | EN –> DIA | Test Set | EN->MSA-> DIA | EN –> DIA |
| madar test lev 0 | 9.54 | 20.25 | madar test nil 2 eg | 8.99 | 19.68 |
| madar test lev 0 lb | 9.03 | 12.10 | arzen | 22.86 | 6.85 |
| LDC test | 4.12 | 6.11 | madar test nil 1 eg | 7.91 | 18.63 |
| madar test lev 1 jo | 8.27 | 16.78 | madar test nil 0 eg | 31.33 | 18.12 |
| **Average** | **7.74** | **13.81** | **Average** | **17.77** | **15.82** |

## A.7 DATA SPLIT

In Table 11, we report the data-split distribution for all countries. We used stratified sampling with subcategory as the stratification variable. The dev and test splits each comprise $\sim$3.76% of the data, yielding about 2K QAs per country per split.

Table 11: Distribution of train, dev, and test splits across countries (two-letter ISO codes). DZ=Algeria, BH=Bahrain, EG=Egypt, IQ=Iraq, JO=Jordan, KW=Kuwait, LB=Lebanon, LY=Libya, MA=Morocco, OM=Oman, PS=Palestine, QA=Qatar, SA=Saudi Arabia, SD=Sudan, SY=Syria, TN=Tunisia, AE=United Arab Emirates, YE=Yemen.

| Split | DZ | BH | EG | IQ | JO | KW | LB | LY | MA | OM | PS | QA | SA | SD | SY | TN | AE | YE | Total |
|---|---|---|---|---|---|---|---|---|---|---|---|---|---|---|---|---|---|---|---|
| Train | 46,503 | 40,414 | 51,982 | 44,594 | 50,658 | 40,123 | 49,962 | 44,576 | 59,843 | 49,925 | 51,101 | 32,178 | 52,419 | 45,680 | 45,683 | 51,020 | 55,166 | 47,805 | 859,632 |
| Dev | 1,931 | 1,938 | 1,964 | 1,938 | 1,938 | 1,934 | 1,944 | 1,938 | 1,945 | 1,949 | 1,913 | 1,950 | 1,943 | 1,941 | 1,939 | 1,947 | 1,935 | 1,936 | 34,923 |
| Test | 1,944 | 1,941 | 1,948 | 1,945 | 1,940 | 1,935 | 1,933 | 1,941 | 1,940 | 1,930 | 1,941 | 1,943 | 1,944 | 1,941 | 1,939 | 1,942 | 1,947 | 1,936 | 34,930 |
| Total | 50,378 | 44,293 | 55,894 | 48,477 | 54,536 | 44,992 | 53,839 | 48,455 | 63,728 | 53,804 | 54,955 | 36,071 | 56,306 | 49,562 | 49,561 | 54,909 | 59,048 | 51,677 | 929,485 |

## B EXPERIMENTAL SETUP

In Table 12, we report the details of the experimental setups for this study, which reports the number of models, modality and language varieties we have experimented with.

## C PROMPTS

### C.1 PROMPT FOR TOPIC & QUERY GENERATION

In Listing 1, we provide the prompts for generating seed topics. We provide the prompts for generating queries for English, MSA, and regional dialects in Listings 2, and 3 Finally, we provide the prompt to generate cultural-relevance scores for seed queries in Listing 4.

```
You are an AI specialized in generating highly relevant topics for image searches based on a
given country, category, and subcategory. Your task is to generate a list of topics that are
highly visual and well-suited for image searches. Ensure that the topics reflect the cultural,
historical, or modern significance of the specified location.
```

Table 12: Evaluated models across input modalities: text ($T$), text+image ($T+I$), speech ($S$), and speech+image ($S+I$), with text as output. A ✓ indicates experiments conducted; ✗ indicates not applicable. **Egy** = Egyptian Arabic (arz), **Lev** = Levantine Arabic (ajp), $Tr$ = transcription, $Tr+I$=transcription+image. FT = Fine-tuning.

| Models | English | | | | | | MSA | | | | | | Egy | | Lev | |
|---|---|---|---|---|---|---|---|---|---|---|---|---|---|---|---|---|
| | T | S | T+I | Tr | Tr+I | S+I | T | S | T+I | Tr | Tr+I | S+I | T | T+I | T | T+I |
| Gemini-pro [T,S,I] | ✓ | ✓ | ✓ | ✓ | ✓ | ✓ | ✓ | ✓ | ✓ | ✓ | ✓ | ✓ | ✓ | ✓ | ✓ | ✓ |
| GPT-4.1 [T,I] | ✓ | ✗ | ✓ | ✓ | ✓ | ✗ | ✓ | ✗ | ✓ | ✓ | ✓ | ✓ | ✓ | ✓ | ✓ | ✓ |
| GPT-4o-audio [S] | ✗ | ✓ | ✗ | ✗ | ✗ | ✗ | ✗ | ✓ | ✗ | ✗ | ✗ | ✗ | ✗ | ✗ | ✗ | ✗ |
| GPT-5 [T,I] | ✓ | ✗ | ✓ | ✓ | ✓ | ✗ | ✓ | ✗ | ✓ | ✓ | ✓ | ✗ | ✓ | ✓ | ✓ | ✓ |
| Phi-4 [T,S,I] | ✓ | ✓ | ✓ | ✓ | ✓ | ✓ | ✓ | ✓ | ✓ | ✓ | ✓ | ✓ | ✓ | ✓ | ✓ | ✓ |
| Qwen-2.5 3B [T,S,I] | ✓ | ✓ | ✓ | ✓ | ✓ | ✓ | ✓ | ✓ | ✓ | ✓ | ✓ | ✓ | ✓ | ✓ | ✓ | ✓ |
| Qwen-2.5 7B [T,S,I] | ✓ | ✓ | ✓ | ✓ | ✓ | ✓ | ✓ | ✓ | ✓ | ✓ | ✓ | ✓ | ✓ | ✓ | ✓ | ✓ |
| FT–Qwen-2.5 3B [T,S,I] | ✓ | ✓ | ✓ | ✓ | ✓ | ✓ | ✓ | ✓ | ✓ | ✓ | ✓ | ✓ | ✓ | ✓ | ✓ | ✓ |

```
Guidelines:
1. Topics should be engaging, highly visual, and unique to the specified country.
2. Ensure a mix of historical, modern, and futuristic aspects based on the subcategory.
3. Use well-known landmarks, cultural elements, or emerging trends where relevant.
4. Prioritize topics that are frequently searched for in image search engines.
5. If the subcategory is broad, ensure a diverse selection covering different aspects.
6. Do not include generic topics that could apply to any country; make them location-specific.
7. If the subcategory is too narrow and lacks visual topics, expand the scope slightly to
include related themes.
8. Generate exactly 10 topics per request. If necessary, include related visual aspects.
9. Avoid redundant or overly generic suggestions.
10. Ensure diversity in the topics; avoid generating closely related topics.

JSON Format:
- Provide a list of topics, each being short, clear, and descriptive (e.g., 'Futuristic
Skyscrapers of Dubai' or 'Traditional Wooden Temples of Japan').

    json
    [
        "Futuristic Skyscrapers",
        "Traditional Mosque"
    ]

Generate **exactly** 10 highly relevant topics for image search based on the following:

- Country: {country}
- Category: {category}
- Subcategory: {subcategory}

If there are fewer than 10 highly relevant topics, expand the scope slightly to related visual
themes.

Ensure the topics are visually engaging, related to the specified country, and match common
image search behavior.
The topics should cover a mix of historical, modern, and futuristic elements unique to the
location.
```

Listing 1: Prompt for generating 10 seed topics for each *(country, category)* pair.

```
You are an expert at generating **highly relevant, human-like image search queries** optimized
for **Google Image Search**.

Your task is to generate **50 unique search queries** that reflect **natural human behavior**,
including:
- **Typos, slang, informal expressions**, and **incomplete or autocomplete-style phrases**.
- Use **descriptive visual terms** such as "HD," "4K," "wallpaper," "real photo," "aesthetic,"
"close-up," "latest pics," etc.
- Mimic **real-world search styles**, including:
- Pure **keywords**
- **Questions** (e.g., "what do [topic] look like")
- **Autocomplete-like fragments** (e.g., "best pics of...")
- **Trending styles** (e.g., "free download,")

Incorporate **localized and culturally relevant elements** from the country provided,
including:
- Dialects, slang, and spelling variations
- Famous **cities, landmarks**, or **cultural symbols**
- Country-specific visual cues, aesthetics, or references

Queries should be:
- **Short, human-like and natural-sounding** (2-5 words on average)
- **Highly visual** and suitable for image search intent
- Focused on the **topics**, not just the country, category or subcategory
- Always returned in **strict JSON format**:
json
{
    "queries": [
        "query 1",
        "query 2",
        "... up to query 50"
    ]
}

Remember to generate exactly 50 unique queries, ensuring a diverse range of search styles and
incorporating elements specific to the given country. Focus on creating queries that real
users might type when searching for images related to the provided topic.

Generate **50** unique, human-like image search queries** based on below information:

- Country: {country}
- Category: {category}
- Subcategory: {subcategory}
- Topics: {topic}
```

Listing 2: Prompt generate queries in English.

```
You are an expert at generating **highly realistic, human-like image search queries in
Arabic** optimized for **Google Image Search**.

Your task is to generate 50 unique Arabic image search queries based on a given **country**,
**category**, **subcategory**, and **topic**. These queries should reflect **how real people
```

from the Arab world search for images**, using both **Modern Standard Arabic (MSA)** and **country-specific dialects** where appropriate.

Follow these guidelines to generate the queries:

1. Reflect natural human behavior:
   - Use informal phrasing, spelling mistakes, colloquial expressions, and incomplete or autocomplete-style fragments.
   - Vary punctuation, phrasing, and structure - some queries should be formal, others casual or conversational.
   - Mimic how people write queries on their phones or in autocomplete (e.g., أجمل صور المغرب, وين ألقى صور الشوارع).

2. Use visual and search-specific descriptors in Arabic:
   - Words like: صور, خلفيات, تنزيل, تحميل مجاني, أجمل صور, 4K, صور حض, حقيقية, من انستقرام, خلفية جوال.

3. Mimic real-world Arabic search styles:
   - Pure keywords
   - **Keyword-based**: e.g., صور اللبس المغربي, خلفيات الصحراء الجزائرية
   - **Questions**: e.g., أين أجد صور الأسواق القديمة في اليمن؟
   - **Incomplete phrases**: e.g., أجمل صور من عمان, تنزيل صور تقليدية

4. Use localized and culturally relevant language:
   - Include Arabic **city names**, dialectal slang, and popular references from the country (e.g., الجلابية, العقال, لبسة مغربية).
   - Mention famous places or cultural features (e.g., pyramids, mosques, old souks, traditional outfits).
   - Dialects to consider: Egyptian, Gulf, Levantine, Maghrebi - depending on the country.

Ensure the queries:
- Are **short (2-5 words)**, natural-sounding, and visually oriented
- **Focus heavily on the topic**
- Avoid any overly formal, robotic phrasing

Return the results in the following **strict JSON format only**:
```json
{
    "queries": [
        "query 1 in Arabic",
        "query 2 in Arabic",
        "... up to query 50 in Arabic"
    ]
}
```

Generate **50** unique, human-like image search queries in Arabic based on below information:

- Country: {country}
- Category: {category}
- Subcategory: {subcategory}

```
- Topics: {topic}
```

Listing 3: Prompt for generating queries in MSA and dialects.

```
You are an expert in evaluating search query effectiveness for image search. Your task is to
rank image search queries based on their relevance to a given location. Focus on specificity,
uniqueness, and cultural significance when ranking them. Assign a relevance score from 1 to
100 and return the output in JSON format.

Given the following list of **image search queries** related to {location}, **evaluate and
assign a relevance score to every single query**.

Each query must receive a **relevance score from 1 to 100**, where:
- 100 represents the highest relevance.
- Higher scores go to queries highlighting **iconic landmarks, cultural elements, or unique
aspects of {location}**.
- Queries mentioning a **location outside of {location}** should receive a **low relevance
score**.
- **DO NOT skip any query** - every query in the list must be assigned a score.
- Queries are in **Arabic and English** - evaluate both equally.
- Queries those are not related to {location} should receive a very low score.

### **List of Queries:**
json
{json.dumps(query_list, ensure_ascii=False)}

**Expected JSON Output Format:**
json
[
    {{"Q": "Eiffel Tower at sunset", "score": 100}},
    {{"Q": "Paris street art", "score": 90}},
]
```

Listing 4: Prompt for generating cultural-relevance score of search queries. The place-holder represents the list of queries.

## C.2 PROMPT FOR IMAGE DESCRIPTION GENERATION

We provide the prompt that we used for image description generation and image categorization in Listing 5.

```
You are an AI assistant specializing in image analysis, filtering, and categorization for
question-answering (QA) systems. Your task is to **describe, classify, and assess** images
based on their relevance and suitability.
### **1. Image Description**
- Provide a **concise, objective** description of the image.
- Extract readable text (if any) and include it under "extracted_text".

### **2. Image Categorization**
Classify the image into **one of the following categories**:
- **Photograph** - Real-world photo.
- **Illustration** - Hand-drawn or digital artwork (e.g., sketches, comics). Excludes branded
mascots in ads.
```

```
- **Advertisement** - Promotional content with branding, pricing, slogans, or call-to-action
text. Includes banners, flyers, or sponsored content.
- **Screenshot/UI Capture** - Software, websites, or apps.
- **Meme/Text Overlay** - An image with overlaid text, often humorous or social.
- **Chart/Infographic** - A diagram or data visualization, such as an infographic or graph.
- **Other** - Any content that does not fit the above categories.

### **3. Suitability Assessment**
- Determine if the image is **suitable** for a QA system based on:
- **Clarity** (clear, readable, and interpretable).
- **Relevance** (must align with the user-provided **topic** and **subtopic**).
- **Content** (must not contain inappropriate elements).
- Provide a **justification** for the suitability decision.

### **4. Response Format (JSON)**
Return results in **structured JSON format**:
{{
    "description": "<concise image description>",
    "extracted_text": "<text extracted from image (if any)>",
    "image_category": "<category>",
    "status": "<suitable/not_suitable>",
    "reason": "<brief explanation>"
}}

Analyze the given image in the context of **Topic: {category.lower()}** and **Subtopic:
{subcategory.lower()}**.

**Image:** {image}

- **Describe the image.**
- **Extract readable text (if any).**
- **Classify the image into a predefined category.**
- **Assess if it is suitable** for a QA system based on clarity, relevance, and content.
```

Listing 5: Prompt for generating bilingual image description and categorization.

## C.3 PROMPT FOR GENERATING QUESTION-ANSWER

We provide a prompt for generating four cultural question–answer pairs per image, as shown in Listing 5.

```
You are an AI assistant specializing in Visual Question Answering (VQA). Your task is to
analyze the given image and generate high-quality Question-Answer (Q&A) pairs for benchmarking
and training large language models (LLMs).

Follow these guidelines carefully:

1. Types of Q&A Pairs (generate all for each image):
    1. Open-ended: A free-form question with an informative answer based on the image.
    2. Multiple-choice: A question with three plausible options, clearly marking the correct
    answer.
    3. True/False: A question-answer pair that can be answered with 'True' or 'False'.
```

For type 1 and 2 you should generate one QA pair for each. For type 3 you should generate
two QA pairs, one with True and one with False.

2. Semantic Focus:
   - Use the following semantic labels to guide your questions. Match the image content to
   the most relevant labels:
       - Location and Place Identification
       - Scene Interpretation and Context
       - Architectural Features and Functions
       - Cultural Significance and Heritage
       - Traditional Clothing and Attire
       - Tourism and Cultural Activities
       - Event and Activity Type
       - Objects, Animals, and Food Recognition
       - National Symbols and Identity
       - Visual Attributes
       - Recreational Activities and Facilities

3. Cognitive Focus:
   - Ensure a balanced mix of:
       - Knowledge-based questions (requiring factual knowledge related to the image).
       - Common sense-based questions (requiring general reasoning or everyday knowledge to
       answer).
   - Assign a label to each question indicating its cognitive focus (knowledge-based or
   common sense-based).

4. Language:
   - All Q&A pairs must be written in native-sounding English.

5. Question Quality:
   - Ensure the questions are natural, conversational, and human-like.
   - Vary the phrasing and difficulty across the different question types. Questions should
   be engaging and thought-provoking. A mix of simple and complex questions is encouraged.

6. Answer Quality:
   - Answers must be factually correct, clear, concise, and well-structured.
   - Use correct grammar and maintain high readability.

7. Cultural Sensitivity:
   - Avoid stereotypes or cultural misrepresentations.
   - Ensure cultural references are accurate and specific to the image.

8. Context Utilization:
   - Use the provided image description, category, and subcategory to enrich the context
   while formulating the questions.

9. Reasoning:
   - For each Q&A pair, also provide a short explanation justifying why the answer is
   correct. Limit the explanation to less than 100 words.

Strictly follow these instructions to ensure the generated VQA data is of the highest quality
and suitable for model evaluation and fine-tuning.

### **Output Format (JSON):**

```
json
{{
    "open-ended": [
        {{"question_en": "...", "answer_en": "...", "rationale": "...", "cognitive_focus":
        "...","semantic_focus": ["...","..."]}},
    ],
    "multiple-choice": [
        {{"question_en": "...", "options_en": ["..."], "correct_answer_en": "...", "rationale":
        "...","cognitive_focus": "...","semantic_focus": ["...","..."]}},
    ],
    "True/False": [
        {{"question_en": "...", "answer_en": "...", "rationale": "...","cognitive_focus":
        "...""","semantic_focus": ["...","..."]}},
        {{"question_en": "...", "answer_en": "...", "rationale": "...","cognitive_focus":
        "...""","semantic_focus": ["...","..."]}},
    ]
}}

Analyze the given image and generate **question-answer pairs with their rationales for each
type: 1) Open-ended, 2) Multiple-choice, 3. True/False QA pairs**.

**Image:** {image}

Use the following information as an additional context for generating questions:
**Description:** {description}
**Category:** {category}
**Subcategory:** {subcategory}
```

Listing 6: Prompt for generating four cultural question-answer pairs per image.

## D  DETAILS OF THE RESULTS

In Table 13, we report results across models, modalities, and languages for all question types. We also present sample text and text+image results in Table 14 on the full dataset, comparing Gemini-Pro, Qwen-2.5 (Omni-3B), and the fine-tuned Qwen-2.5 (Omni-3B). In Figure 8, we report LLM-Judge score across modalities comparing different models.

## E  QUALITATIVE ANALYSIS

We conducted an error analysis to understand the types of images that models failed to answer. Figure 9 provides examples of images where almost all models failed. These cases suggest that the models may require more contextual or culturally specific information to answer such questions correctly. We also performed category- and subcategory-wise performance analysis across all models. Our findings show that models perform well in some categories (e.g., Heritage & History) but struggle in others (e.g., eSports & Gaming), as presented in Figure 11. In Figure 12, we show category-wise MCQ performance for Egyptian dialects using the Qwen-3B base and fine-tuned models. The results demonstrate that fine-tuning significantly improves performance in several categories (e.g., Objects).

In Figure 13, we report the performance by grouping knowledge vs. commonsense based QA, which are obtained using the Gemini model.

Table 13: Evaluation results across languages and speech modality combinations. **F1** = F1 BERTScore, **Judge** = LLM-as-judge score (GPT-4.1), **Acc** = accuracy, **T** = text, **Tr** = transcription, **T+I** = text+image, **Tr+I** = transcription+image. Judge scores range from 1 to 10. Gemini = Gemini-2.5-pro. The best model across all modalities for Open-Ended Judge and True/False Accuracy, is shown in bold.

| | | English | | | | MSA | | | |
|---|---|---|---|---|---|---|---|---|---|
| | | Open-ended | | TF 1 | TF 2 | Open-ended | | TF 1 | TF 2 |
| Model | Modality | F1 | Judge | Acc | Acc | F1 | Judge | Acc | Acc |
| GPT-4.1 | T | 0.61 | 6.23 | 0.63 | 0.79 | 0.58 | 6.43 | 0.69 | 0.71 |
| | T+I | 0.72 | **8.58** | 0.96 | **0.99** | 0.61 | **8.20** | 0.96 | **0.99** |
| | Tr | 0.55 | 5.08 | 0.28 | 0.89 | 0.77 | 3.55 | 0.32 | 0.82 |
| | Tr+I | 0.69 | 5.34 | 0.61 | 0.96 | 0.79 | 3.36 | 0.81 | 0.93 |
| GPT-4o-audio | S | 0.60 | 5.25 | 0.40 | 0.68 | 0.78 | 3.54 | 0.62 | 0.45 |
| GPT-5 | T | 0.58 | 6.19 | 0.59 | 0.87 | 0.55 | 6.46 | 0.77 | 0.62 |
| | T+I | 0.63 | 8.34 | 0.97 | **0.99** | 0.56 | 8.04 | 0.96 | **0.99** |
| | Tr | 0.53 | 5.07 | 0.30 | 0.88 | 0.73 | 3.55 | 0.33 | 0.78 |
| | Tr+I | 0.60 | 5.16 | 0.63 | 0.96 | 0.74 | 3.58 | 0.83 | 0.93 |
| Qwen2.5-7B | T | 0.54 | 5.08 | 0.57 | 0.71 | 0.52 | 4.43 | 0.77 | 0.44 |
| | T + I | 0.60 | 5.06 | 0.97 | 0.98 | 0.53 | 4.46 | 0.96 | 0.94 |
| | Tr | 0.94 | 3.90 | 0.33 | 0.74 | 0.89 | 3.23 | 0.51 | 0.56 |
| | Tr + I | 0.70 | 5.98 | 0.69 | 0.94 | 0.45 | 3.39 | 0.23 | 0.83 |
| | S | 0.53 | 4.00 | 0.37 | 0.73 | 0.73 | 2.99 | 0.56 | 0.56 |
| | S + I | 0.57 | 5.87 | 0.88 | 0.45 | 0.73 | 4.38 | 0.83 | 0.47 |
| Gemini-2.5-pro | T | 0.55 | 5.24 | 0.81 | 0.71 | 0.53 | 5.62 | 0.71 | 0.78 |
| | T+I | 0.60 | 6.64 | 0.93 | 0.98 | 0.54 | 6.59 | 0.94 | 0.97 |
| | Tr | 0.52 | 4.36 | 0.39 | 0.78 | 0.25 | 4.70 | 0.32 | 0.87 |
| | Tr+I | 0.57 | 6.29 | 0.58 | 0.94 | 0.25 | 6.22 | 0.85 | 0.93 |
| | S | 0.47 | 3.79 | 0.39 | 0.69 | 0.24 | 3.46 | 0.36 | 0.73 |
| | S+I | 0.64 | 6.48 | 0.94 | 0.44 | 0.25 | 5.89 | 0.94 | 0.53 |
| Phi-4 | T | 0.51 | 4.77 | 0.39 | 0.83 | 0.49 | 3.52 | 0.60 | 0.59 |
| | T + I | 0.56 | 5.88 | 0.82 | 0.93 | 0.50 | 3.95 | 0.89 | 0.61 |
| | Tr | 0.49 | 3.81 | 0.31 | 0.78 | 0.71 | 2.76 | 0.39 | 0.70 |
| | Tr + I | 0.54 | 5.29 | 0.62 | 0.87 | 0.67 | 2.39 | 0.34 | 0.74 |
| Qwen2.5-3B | T | 0.53 | 4.79 | 0.53 | 0.78 | 0.52 | 3.87 | 0.82 | 0.40 |
| | T+I | 0.48 | 5.10 | 0.95 | 0.97 | 0.51 | 4.79 | 0.97 | 0.87 |
| | Tr | 0.48 | 3.88 | 0.35 | 0.79 | 0.73 | 2.93 | 0.80 | 0.29 |
| | Tr+I | 0.40 | 3.74 | 0.69 | 0.94 | 0.72 | 2.84 | 0.36 | 0.72 |
| | S | 0.39 | 2.96 | 0.36 | 0.73 | 0.69 | 2.59 | 0.71 | 0.35 |
| | S+I | 0.40 | 3.53 | 0.85 | 0.43 | 0.69 | 3.22 | 0.85 | 0.33 |
| FT (Qwen-2.5-3B) | T | 0.73 | 6.39 | 0.92 | 0.86 | 0.64 | 5.85 | 0.89 | 0.84 |
| | T + I | 0.78 | 8.29 | **0.98** | **0.99** | 0.68 | 7.47 | **0.98** | 0.98 |
| | Tr | 0.66 | 5.06 | 0.82 | 0.82 | 0.79 | 4.90 | 0.82 | 0.71 |
| | Tr + I | 0.74 | 7.76 | 0.85 | 0.96 | 0.79 | 4.90 | 0.84 | 0.70 |
| | S | 0.69 | 5.30 | 0.89 | 0.42 | 0.80 | 5.19 | 0.82 | 0.46 |
| | S + I | 0.77 | 8.13 | 0.92 | 0.43 | 0.83 | 7.07 | 0.90 | 0.50 |

## F   RELATED WORK

Table 15 provides a comparative overview of existing multimodal and multilingual benchmarks. It summarizes each benchmark's supported modalities (text, image, speech), multilingual coverage, number of language varieties and scripts, domains, dataset size, question types and forms, and annotation methods. The table highlights differences in scale, linguistic diversity, and task design across benchmarks, illustrating where *OASIS* fits in terms of multimodality, multilingual support, dataset size, and question diversity.

Table 14: **Open-ended** LLM-as-Judge results (MSA on top, EN below) on the full dataset. **Country codes** are shown as columns: DZ (Algeria), BH (Bahrain), EG (Egypt), IQ (Iraq), JO (Jordan), KW (Kuwait), LB (Lebanon), LY (Libya), MA (Morocco), OM (Oman), PS (Palestine), QA (Qatar), SA (Saudi Arabia), SD (Sudan), SY (Syria), TN (Tunisia), AE (UAE), YE (Yemen). **Model settings**: T = Text, T+I = Text+Image.

| | DZ | BH | EG | IQ | JO | KW | LB | LY | MA | OM | PS | QA | SA | SD | SY | TN | AE | YE | **Avg.** |
|---|---|---|---|---|---|---|---|---|---|---|---|---|---|---|---|---|---|---|---|
| **English** | | | | | | | | | | | | | | | | | | | |
| Gemini-Pro (T) | 5.60 | 5.78 | 5.78 | 5.40 | 5.74 | 5.64 | 5.62 | 5.53 | 5.74 | 5.85 | 5.63 | 5.78 | 5.88 | 5.48 | 5.61 | 5.58 | 6.04 | 5.68 | **5.69** |
| Gemini-Pro (T+I) | 7.00 | 7.01 | 6.61 | 6.80 | 6.73 | 6.97 | 7.00 | 6.91 | 6.96 | 6.96 | 6.82 | 6.90 | 6.80 | 6.90 | 6.94 | 6.96 | 6.98 | 6.98 | **6.90** |
| Qwen2.5 (T) | 3.74 | 3.81 | 3.96 | 3.89 | 3.86 | 3.79 | 3.84 | 3.77 | 3.86 | 3.98 | 3.92 | 3.88 | 3.87 | 3.84 | 3.83 | 3.71 | 3.90 | 3.77 | **3.84** |
| Qwen2.5 (T+I) | 4.86 | 4.91 | 4.92 | 5.00 | 4.84 | 4.90 | 4.89 | 4.83 | 4.95 | 5.00 | 5.03 | 4.95 | 4.99 | 4.94 | 4.78 | 4.74 | 4.97 | 4.90 | **4.91** |
| Qwen2.5-FT (T) | 5.78 | 5.89 | 5.88 | 5.72 | 5.89 | 5.63 | 5.71 | 5.76 | 6.07 | 6.10 | 5.81 | 6.05 | 5.92 | 5.66 | 5.82 | 5.79 | 6.07 | 5.75 | **5.85** |
| Qwen2.5-FT (T+I) | 7.48 | 7.41 | 7.50 | 7.43 | 7.53 | 7.42 | 7.45 | 7.46 | 7.58 | 7.63 | 7.51 | 7.46 | 7.51 | 7.37 | 7.44 | 7.40 | 7.54 | 7.37 | **7.47** |
| **MSA** | | | | | | | | | | | | | | | | | | | |
| Gemini-Pro (T) | 5.44 | 5.59 | 5.55 | 5.28 | 5.54 | 5.37 | 5.50 | 5.42 | 5.53 | 5.65 | 5.46 | 5.56 | 5.65 | 5.35 | 5.45 | 5.37 | 5.69 | 5.53 | **5.50** |
| Gemini-Pro (T+I) | 7.73 | 7.20 | 6.84 | 7.00 | 6.95 | 7.23 | 7.25 | 7.19 | 7.12 | 7.20 | 7.04 | 7.01 | 6.94 | 7.14 | 7.15 | 7.23 | 7.17 | 7.20 | **7.14** |
| Qwen2.5 (T) | 4.70 | 4.82 | 4.94 | 4.74 | 4.94 | 4.58 | 4.80 | 4.71 | 4.77 | 4.87 | 4.90 | 4.78 | 4.82 | 4.73 | 4.70 | 4.71 | 4.79 | 4.74 | **4.78** |
| Qwen2.5 (T+I) | 5.16 | 5.30 | 5.39 | 5.27 | 5.41 | 5.14 | 5.30 | 5.20 | 5.29 | 5.31 | 5.39 | 5.14 | 5.34 | 5.21 | 5.25 | 5.27 | 5.29 | 5.23 | **5.27** |
| Qwen2.5-FT (T) | 6.39 | 6.41 | 6.49 | 6.25 | 6.42 | 6.20 | 6.30 | 6.27 | 6.47 | 6.62 | 6.44 | 6.52 | 6.53 | 6.17 | 6.38 | 6.34 | 6.58 | 6.31 | **6.40** |
| Qwen2.5-FT (T+I) | 8.29 | 8.35 | 8.36 | 8.24 | 8.33 | 8.32 | 8.30 | 8.29 | 8.36 | 8.36 | 8.31 | 8.23 | 8.34 | 8.17 | 8.18 | 8.27 | 8.27 | 8.24 | **8.29** |

Table 15: Comparison of multimodal and multilingual benchmarks. **Mod**: Modalities (Text = T, Image = I, Speech = S). **Multi**: Multilingual support. **Lang**: # of languages varieties. **Script**: # of writing scripts. **Dom**: # of domains. **Samp**: Total samples. **QTypes**: Question types (MCQ = multiple-choice, SVQA = short visual QA, LVQA = long visual QA, TF = true/false, OE = open-ended, Y/N = yes/no). **QForms**: Question forms (Fixed or Diverse). **Annot**: Annotation type (Auto = automatic, Manual = human, Auto+Manual = hybrid). * 10K QA pairs associated with 5,239 images. [†] 1,999 QA pairs associated with 515 images.

| Benchmark | Mod | Multi | Lang. | Script | Dom | Samp | QTypes | QForms | Annot |
|---|---|---|---|---|---|---|---|---|---|
| CVQA (Romero et al., 2025) | T,I | ✓ | 31 | 13 | 10 | 5,239* | MCQ | Fixed | Manual |
| ALM-Bench (Vayani et al., 2025) | T,I | ✓ | 100 | 24 | 19 | 22,763 | MCQ, SVQA, LVQA, TF | Diverse | Auto+Manual |
| CulturalVQA (Nayak et al., 2024) | T,I | ✓ | 1 | 1 | 5 | 2,378 | SVQA | - | Manual |
| SeaVQA (Urailertprasert et al., 2024) | T,I | ✓ | 1 | 1 | - | 515[†] | MCQ | - | Manual |
| Camel-Bench (Ghaboura et al., 2025) | T,I | ✓ | 2 | 2 | 5 | 29,036 | SVQA, LVQA | Diverse | Auto+Manual |
| MM-Vet (Yu et al., 2024) | T,I | ✗ | 1 | 1 | 16 | 218 | SVQA, LVQA | Fixed | Manual |
| Pangea-Bench (Yue et al.) | T,I | ✓ | 47 | 13 | 18 | - | MCQ, SVQA | Fixed | Auto |
| MMBench (Liu et al., 2024) | T,I | ✓ | 2 | 2 | 20 | 3,217 | MCQ | Fixed | Manual |
| MaRVL (Collini et al., 2025) | T,I | ✓ | 5 | 3 | 11 | 5,670 | TF | Fixed | Manual |
| M3Exam (Zhang et al., 2023) | T,I | ✓ | 9 | 3 | 4 | 12,317 | MCQ | Diverse | - |
| xGQA (Pfeiffer et al., 2022) | T,I | ✓ | 8 | 5 | - | 12,578 | Y/N, SVQA | Fixed | - |
| OmniBench (Li et al., 2024b) | T,I,S | ✓ | 2 | 2 | 8 | 1,142 | MCQ | Diverse | Manual |
| Pearl (Alwajih et al., 2025) | T,I | ✗ | 1 | 1 | 10 | 309,000 | 13 types | Diverse | Auto+Manual |
| *OASIS* | T,I,S | ✓ | 2 | 4 | 31 | 0.92M | OE, MCQ, TF | Diverse | Auto+Manual |

# G    ANNOTATION GUIDELINES

This section provides the annotation guidelines used for *(i)* voice recording and *(ii)* quality assessment of QA pairs. These instructions were shown to annotators during data collection to ensure consistency, clarity, and high-quality annotations across countries and languages.

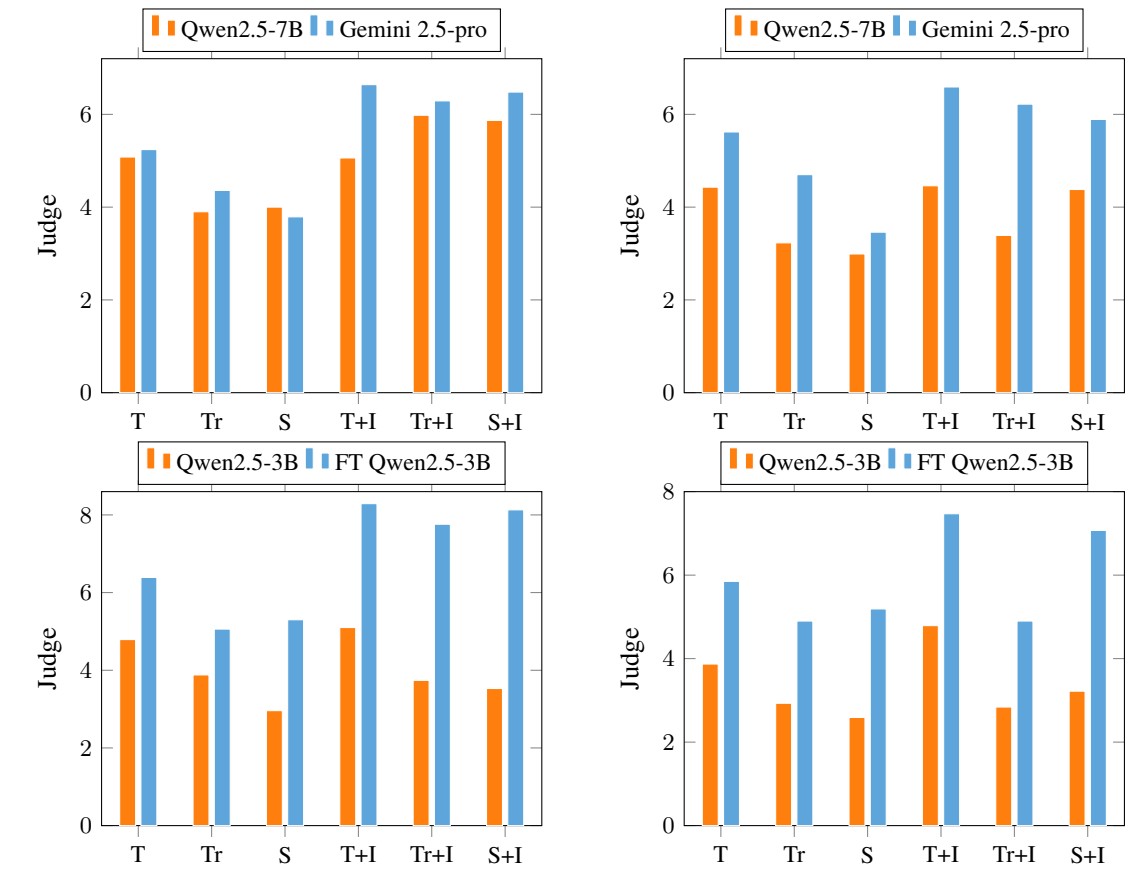

Figure 8: LLM-Judge scores across modalities. Top row: Qwen2.5-7B vs Gemini 2.5-pro for English (left) and MSA (right). Bottom row: Qwen2.5-3B vs its fine-tuned variant for English (left) and MSA (right).

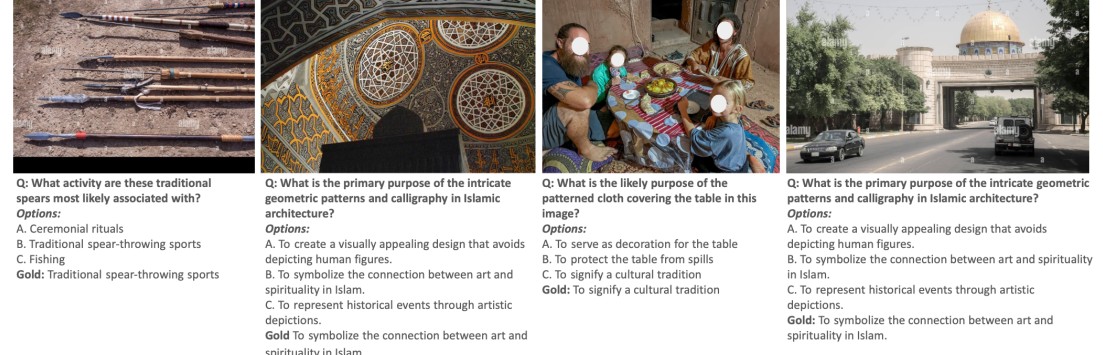

Figure 9: Examples of images where models incorrectly answered.

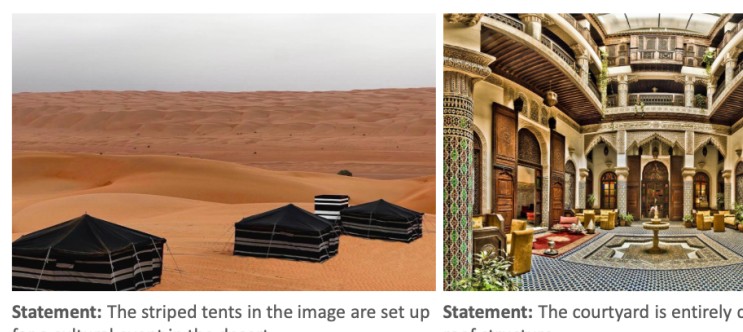

**Statement:** The striped tents in the image are set up for a cultural event in the desert.
**Gold:** True

**Statement:** The courtyard is entirely covered by a roof structure.
**Gold:** True

Figure 10: Examples of images where the fine-tuned model answered correctly.

## G.1 QA ANNOTATION GUIDELINES

Annotators evaluated image-based question–answer (QA) pairs including one open-ended question, one multiple-choice question, and two true/false items.

On the annotation interface, you will see the following:
1. An image
2. A description of the image to help you understand the image better
3. The different types of questions, answers, and a rationale for each answer, all related to the image shown

The types of questions will be:
- One open-ended question
- One multiple-choice question with choices
- Two true/false questions with the selected answer

### G.1.1 ANNOTATION TASK

- Decide if the image and its associated questions are related to the specified location.
- For each type of question, score the clarity and quality on a scale from 1–5.
- Indicate whether answering the question requires external knowledge (e.g., searching online or consulting information not present in the image).
- For each answer, score the correctness (1–5 for open-ended and multiple-choice; 1–3 for true/false).
- For each rationale, score the quality: Clarity, Informativeness, Plausibility, Faithfulness (1–5).
- If any score is less than 4, choose a reason for revision.

### G.1.2 SCORING QUESTIONS AND ANSWERS

**Open-Ended Question and its Answer**

**Question Quality (1–5):** Assess clarity, relevance, and lack of ambiguity. Revision reasons (if score < 4):
- Unclear or ambiguous
- Not relevant to the image
- Hard to understand
- Requires external knowledge

**Answer Quality (1–5):** Assess factuality, conciseness, and grounding in the image. Revision reasons:
- Incorrect or unsupported by the image

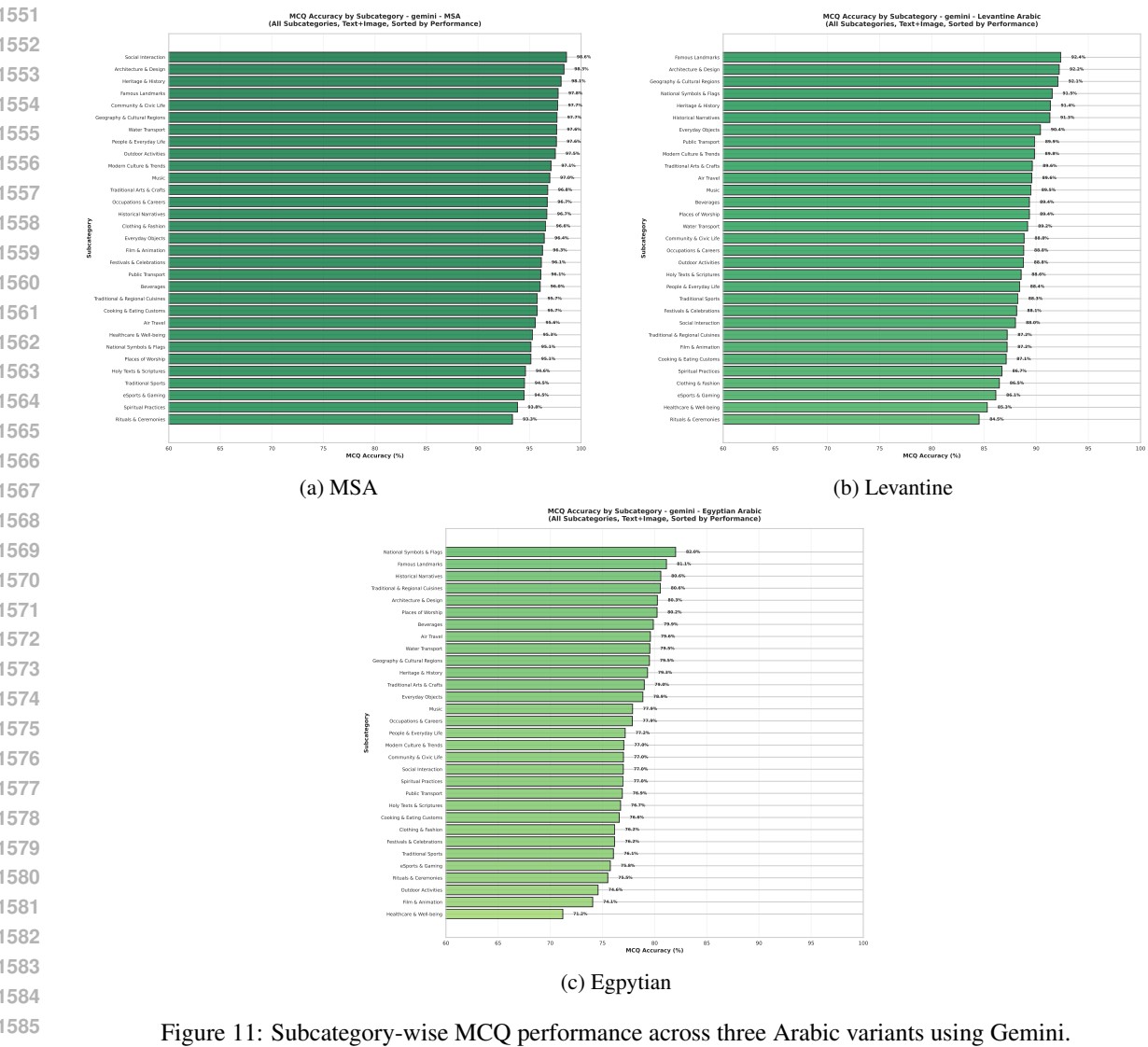

Figure 11: Subcategory-wise MCQ performance across three Arabic variants using Gemini.

- Incomplete or missing key information
- Speculative or assumption-based

**Rationale Quality:**
- Clarity & Informativeness (1–5)
- Plausibility & Faithfulness (1–5)

**Multiple-Choice Question**

**Question Quality (1–5):** Clarity, specificity, relevance. Revision reasons (if score < 4):
- Unclear or ambiguous
- Not relevant to the image
- Hard to understand

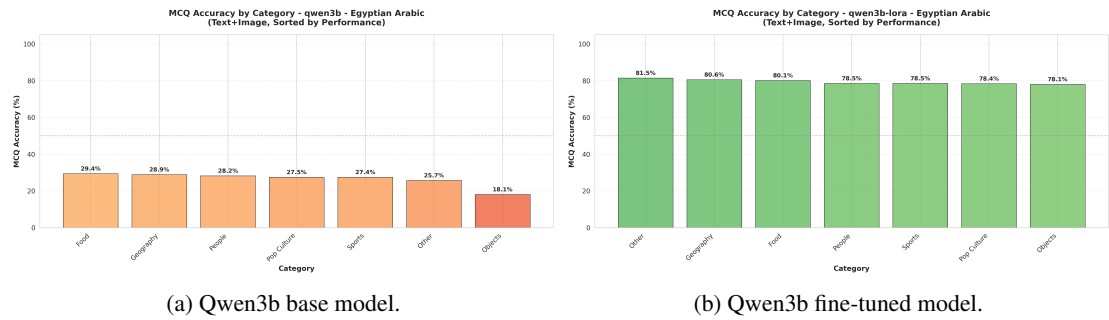

(a) Qwen3b base model.

(b) Qwen3b fine-tuned model.

Figure 12: Category-wise MCQ performance on Egyptian dialects with and without the fine-tuned Qwen-3B model.

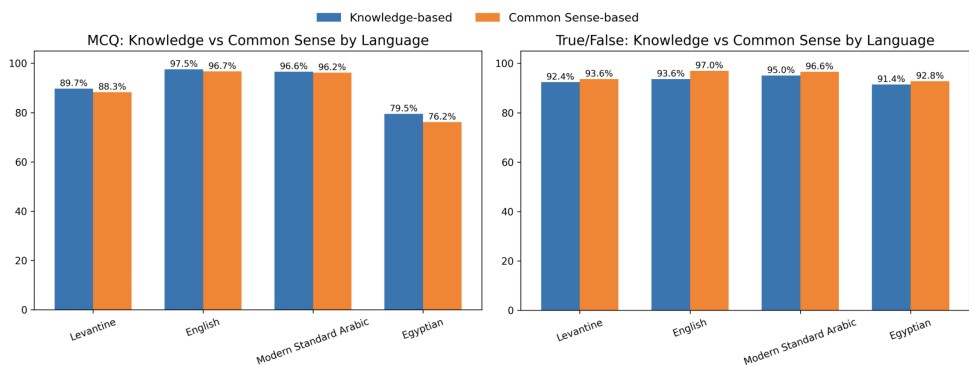

Figure 13: Comparison of *knowledge* vs. *commonsense* MCQ performance of the Gemini model across different language variants.

**Answer Quality (1–5):** Correctness and image support. Revision reasons:
- Options overlap in meaning
- Irrelevant or implausible options
- Vague or confusing options

**Rationale Quality:**
- Clarity & Informativeness (1–5)
- Plausibility & Faithfulness (1–5)

**True/False Questions and Selected Answer**

**Statement Quality (1–5):** Clarity, factual nature, verifiability from the image. Revision reasons (if score < 4):
- Unclear or ambiguous
- Not factual

**Selected Answer Quality (1–3):** Correctness and factual grounding. Revision reasons:
- Incorrect or unsupported by the image
- Incomplete or missing information
- Speculative or assumption-based

**Rationale Quality:**
- Clarity & Informativeness (1–5)

- Plausibility & Faithfulness (1–5)

**General Annotation Principles**

- Avoid speculation beyond what is visible or inferable from the image.
- Mark questions requiring external knowledge.
- Always select a revision reason for scores below 4.
- You may enlarge the image by opening it in a new window.

### G.2 VOICE RECORDING INSTRUCTIONS

Annotators were asked to record spoken the question The following guidelines were displayed in the interface:

- **Read the Sentence:** A sentence appears on the screen. Read it aloud clearly.
- **Record:** Click *Record* to start capturing your voice. Speak naturally and clearly.
- **Playback:** After finishing, click *Play* to listen to your recording.
- **Review:** If satisfied with the audio, click *Submit* to save it.
- **Re-record if Needed:** If the recording is unclear or incorrect, click *Delete* and rerecord the sentence.
- **Submit:** Once satisfied, click *Submit* to store the final version and proceed.

## H DATASET SAMPLE EXAMPLES

In this section, we provide example images, as shown in Figures 14 and 15, along with their associated metadata, language variants, and corresponding open-ended questions.

1692
1693
1694
1695
1696
1697
1698
1699
1700
1701
1702
1703
1704
1705
1706
1707
1708
1709
1710
1711
1712
1713
1714
1715
1716
1717
1718
1719
1720
1721
1722
1723
1724
1725
1726
1727
1728
1729
1730
1731
1732
1733
1734
1735
1736
1737
1738

**Image Description:**

**MSA:** "شخصان يرتديان الزي التقليدي ويقفان على خيول مزينة خلال حدث في الهواء الطلق. وتظهر في خلفية الفيديو لافتة تحمل نصاً باللغة العربية."

**Eng:** "Two individuals dressed in traditional attire standing on decorated horses during an outdoor event. A banner with Arabic text is visible in the background."

**ARZ:** "اتنين لابسين لبس تقليدي واقفين على خيول مزينة في حفلة في الهواء الطلق. في يافطة وراهم مكتوب عليها بالعربي",

**AJP:** "شخصين لابسين لباس تقليدي واقفين على خيول مزينة خلال فعالية خارجية. في لافتة مكتوب عليها بالعربي بابنة بالخلفية."

**Category:** *"Sports & Recreation"*,
**Subcategory:** *"Traditional Sports"*,
**Country:** *Oman*

**Open-ended:**
Question: *(What type of event is depicted in this image, and what is its cultural significance?)*

**Answer**: *The image shows a traditional equestrian event, likely showcasing cultural heritage through horse riding and traditional attire. Such events are significant as they preserve and celebrate historical practices and traditions.*

---

**Image Description:**

**MSA:** "تعرض الصورة الملابس الفلسطينية التقليدية، وتحديداً ثوباً مطرزاً بأشكال معقدة باللون الأحمر والأرجواني والبيج. ويشمل الثوب أكمام طويلة، شرابات، وغطاء الرأس مطابقة. تتميز الخلفية بوسائد ذات تصميم تطريز مماثلة.",

**Eng:** "The image showcases traditional Palestinian clothing, specifically a richly embroidered thobe with intricate patterns in red, purple, and beige tones. The garment includes long sleeves, tassels, and a matching headscarf. The background features cushions with similar embroidery designs."

**ARZ:** "الصورة بتوّري لبس فلسطيني تقليدي، تحديداً ثوب مطرّز تطريز تقيل بالوان أحمر وبنفسجي وبيج. اللبس فيه أكمام طويلة وشراشيب وكمان طرحة ماشية معاه. في الخلفية فيه مخدات عليها تطريز شبه اللي على اللبس.",

**AJP:** "الصورة بتعرض اللباس الفلسطيني التقليدي، وبالتحديد ثوب مطرز بشكل غني بنقشات معقدة بالوان أحمر، بنفسجي وبيج. اللبس فيه أكمام طويلة، شراشيب، وطرحة رأس مطابقة. بالخلفية في مخدات عليها تطريزات مشابهة."

**Category:** *"Sports & Recreation"*,
**Subcategory:** *"Traditional Sports"*,
**Country:** *Palestine*

---

**True/False:**
Question: *(The garment shown in the image is a traditional Palestinian thobe)*

**Answer**: *True*
Rationale: *The image clearly depicts a traditional Palestinian thobe, identifiable by its intricate embroidery and cultural design elements.*

---

Question: *(The embroidery patterns on the thobe are identical across all Palestinian regions.)*

**Answer**: *False*
Rationale: *Embroidery patterns on Palestinian thobes vary by region, with each design representing specific local traditions and identities.*

---

**Image Description:**

**MSA:** "صورة فوتوغرافية لهرم الجيزة الأكبر، الموجود في مصر. و يحيط بالهرم رمال الصحراء وهياكل أصغر، مع سماء زرقاء واضحة في الخلفية.",

**Eng:** "A photograph of the Great Pyramid of Giza, located in Egypt. The pyramid is surrounded by desert sand and smaller structures, with a clear blue sky in the background."

**ARZ:** "صورة لهرم خفرع في الجيزة، في مصر. الهرم حواليه رملة صحرا وبُنَر صغيرة، والسماء وراهم صافية وزرقا.",

**AJP:** "صورة لهرم الجيزة الأكبر الموجود بمصر. الهرم محاط برمال الصحراء وبنايات أصغر، والسماء ورا واضحة وزرقا."

**Category:** *"Geography, Buildings & Landmarks"*,
**Subcategory:** *"Architecture & Design"*,
**Country:** *Egypt*

---

**Multiple Choice:**
Question: *(Which material was primarily used to construct the Great Pyramid of Giza?)*

1. **Limestone**
2. Granite
3. Sandstone

---

**Image Description:**

**MSA:** "تظهر الصورة قوارب خشبية تقليدية راسية في ميناء مع أفق المدينة في الخلفية ، ويتميز بناطحات السحاب والمباني الحديثة. القوارب مجهزة بمعدات الصيد والإمدادات.",

**Eng:** "The image depicts traditional wooden boats docked at a harbor with a city skyline in the background, featuring modern skyscrapers and buildings. The boats are equipped with fishing gear and supplies."

**ARZ:** "الصورة بتوّري مراكب خشب تقليدية راكنة في المينا ، ووراهم منظر المدينة اللي فيها أبراج عالية ومباني حديثة. المراكب عليها عدة صيد وحاجات تانية",

**AJP:** "الصورة بتبين قوارب خشب تقليدية راسية بالميناء، وخلفيتها أفق المدينة اللي فيها ناطحات سحاب ومباني حديثة. القوارب مجهزة بأدوات صيد ومؤن."

**Category:** *"Vehicles & Transportation"*,
**Subcategory:** *"Public Transport"*,
**Country:** *Kuwait*

---

**Open-ended:**
Question: *(The image shows a rural village with traditional houses in the background.)*

**Answer**: *The image depicts traditional wooden fishing boats, commonly known as dhows, which are equipped with fishing gear and used for fishing purposes.*

Figure 14: Examples.

1739
1740
1741
1742
1743
1744
1745
1746
1747
1748
1749
1750
1751
1752
1753
1754
1755
1756
1757
1758
1759
1760
1761
1762
1763
1764
1765
1766
1767
1768
1769
1770
1771
1772
1773
1774
1775
1776
1777
1778
1779
1780
1781
1782
1783
1784
1785

**Metadata**

**Image Description:**
**MSA:** "طبق من ورق العنب المحشو (دولما) مزين بشرائح الليمون والشبت الطازج، موضوع على سطح رمادي مع شوكة وسكين في مكان قريب، ويمكن رؤية قطعة قماش بيج في الخلفية.",
**Eng:** "A plate of stuffed grape leaves (dolmas) garnished with lemon slices and fresh dill, placed on a gray surface with a fork and knife nearby. A beige cloth is visible in the background.",
**ARZ:** طبق ورق عنب محشي عليه شرايح لمون وشبت فريش، محطوط على سطح رمادي وجنبه شوكة وسكينة. في قماشة بيج بايتة في الخلفية.",
**AJP:** صحن ورق عنب محشي مزين بشرائح ليمون وشبت طازة، محطوط على سطح رمادي وفي شوكة وسكينة جنب الصحن. في قماشة بيج بايتة بالخلفية.",

**Category:** *"Food & Cooking",*
**Subcategory:** *"Traditional & Regional Cuisines",*
**Country:** *Iraq*

**Open-ended:**
Question: *(What dish is shown in the image, and what are its key ingredients?)*

*Answer: The dish shown is stuffed grape leaves, also known as dolmas. Key ingredients typically include grape leaves, rice, and a variety of seasonings, and it is often garnished with lemon slices and fresh dill.*

**Metadata**

**Image Description:**
**MSA:** "صورة لتمثال تاريخي يضم ثلاثة أشكال، مع علامات واضحة للضرر على التمثال. في الخلفية، هناك مسجد مع المآذن والقباب متعددة، جنبا إلى جنب مع المباني الحديثة.",
**Eng:** "A photograph of a historical statue featuring three figures, with visible damage marks on the statue. In the background, there is a mosque with multiple minarets and domes, along with modern buildings."
**ARZ:** "صورة لتمثال تاريخي فيه تلات اشخاص، والتمثال باين عليه علامات تكسير. في الخلفية فيه جامع كبير ليه كذا مئذنة وقباب، ومعاهم مباني حديثة.",
**AJP:** "صورة لتمثال تاريخي يضم تلات شخصيات، وعليه علامات ضرر واضحة. بالخلفية في جامع فيه أكتر من مئذنة وقبة، ومعه مباني حديثة.",

**Category:** *"History, Geography & National Identity",*
**Subcategory:** *"Historical Narratives",*
**Country:** *Lebanon*

**Open-ended:**
Question: *(What is the significance of the damaged statue shown in the image, and what historical narrative might it represent?)*

*Answer: The statue represents the Martyrs' Monument in Beirut, commemorating Lebanese nationalists executed by the Ottoman Empire in 1916. The damage symbolizes the turbulent history of Lebanon, including wars and conflicts.*

**Metadata**

**Image Description:**
**MSA:** "شخصان يعملان على فن فسيفساء البلاط في ورشة عمل، محاطة بالأدوات والبلاط الملون والمواد الحجرية."
**Eng:** "Two individuals working on mosaic tile art in a workshop setting, surrounded by tools, colorful tiles, and stone materials."
**ARZ:** "اتنين شغالين على فن الفسيفساء في ورشة، حواليهم أدوات وبلاط ملون وحجارة."
**AJP:** "شخصين عم يشتغلوا على فن الفسيفساء بورشة، وحواليهم أدوات وبلاط ملون ومواد حجرية.",

**Category:** *"People, Society & Education",*
**Subcategory:** *"Occupations & Careers",*
**Country:** *Morocco*

**Open-ended:**
Question: *(What type of art are the individuals creating in this workshop?)*

*Answer: The individuals are creating mosaic tile art, which involves arranging small, colorful tiles to form patterns or images.*

**Metadata**

**Image Description:**
**MSA:** "منظر جوي للكعبة في مكة المكرمة محاطة بالمصلين ورافعات البناء، مع منطقة المسجد المحيطة والبنية التحتية المرئية.",
**Eng:** "Aerial view of the Kaaba in Mecca surrounded by worshippers and construction cranes, with the surrounding mosque area and infrastructure visible.",
**ARZ:** "صورة من فوق الكعبة في مكة وسط الناس اللي بيصلوا حواليها وفيه أوناش بناء، والمنطقة اللي حواليها من الجامع والبنية التحتية باينة.",
**AJP:** "منظر جوي للكعبة في مكة محاطة بالمصلين والرافعات، مع ظهور منطقة المسجد والبنية التحتية الحديثة.",

**Category:** *"Religion & Spirituality",*
**Subcategory:** *"Spiritual Practices",*
**Country:** *Saudi Arabia*

**True/False:**
Question: *(The Kaaba is located in the city of Mecca.)*

*Answer: True*

Question: *(The Kaaba is primarily used for architectural exhibitions.)*

*Answer: False*

**Metadata**

**Image Description:**
**MSA:** "مجموعة من الأشخاص يرتدون ملابس تقليدية يشاركون في حدث ثقافي مع الطبول والتلويح بالعلم في مكان خارجي يتميز بالعمارة الشرق أوسطية والسجاد المنقوش.",
**Eng:** "Group of people in traditional attire participating in a cultural event with drumming and flag waving in an outdoor setting featuring Middle Eastern architecture and patterned carpets."
**ARZ:** ""مجموعة ناس لابسين لبس تقليدي يشاركون في احتفال ثقافي، فيه طبول وأعلام بيرفرفوا، وكل ده في مكان مفتوح فيه عمارة على الطراز الشرقي وسجاجيد منقوشة.",
**AJP:** "مجموعة من الناس بلباس تقليدي بفعالية ثقافية فيها قرع طبول وتلويح بالأعلام بمكان خارجي فيه عمارة شرق أوسطية وسجاد مزخرف."

**Category:** *"Culture, Arts & Entertainment",*
**Subcategory:** *"Heritage & History",*
**Country:** *Qatar*

**Multiple-Choice:**
Question: *(Which country's national flag is prominently displayed in the image?)*

1. **Qatar**
2. Oman
3. Kuwait

Figure 15: Examples.

