# OpenReview forum: "EverydayMMQA: A Multimodal and Multilingual Framework for Culturally Grounded VQA"
_ICLR.cc/2026/Conference — Submitted to ICLR 2026_

### Official Review · Reviewer_Y1YF · 2025-10-29

**Soundness:** 1
**Presentation:** 2
**Contribution:** 3
**Rating:** 2
**Confidence:** 4

**Summary:**

This paper proposes a new benchmark and framework for generating multilingual and culturally relevant multimodal QA. The resulting dataset, OASIS, covers 18 countries and multiple modalities. The authors also evaluate several existing models on this benchmark.

**Strengths:**

The motivation is strong and addresses an important gap in multimodal and multilingual QA research. The resulting dataset is large-scale and has the potential to be highly beneficial for the research community.

**Weaknesses:**

I’m concerned about the direction of this data. If the aim is to have a multimodal QA mainly focusing on visual, then we should expect the model to get ~0 performance if the question is only given in text. i.e., there is no image so you can’t really answer the question (as mentioned in l36 and the example in Figure 1)

But, we see non-zero text-only performance, which means some questions are solvable without the image anyway, hence making this data not necessarily VQA. If the image is supposed to be an additional modality that can be utilized but not mandatory for answering, then this should be stated more explicitly without contradicting the examples (Figure 1). This should also be explicitly designed both in QA generation and human validation.

The implication on the improved result for T+I modality perhaps is not because of the “language burden” as stated. The boost may simply be because some questions require the image to be answered (again, as in Figure 1).

Another concern is on the LLM-generated data for benchmarking. The data is validated somehow (i.e., Likert scale), but it is not detailed enough, especially for machine-generated data. The Likert scale is not clear on what it measures (question difficulty? question quality? what is the definition of “quality”?). Providing the annotation guideline, including the exact Likert scale used, would be useful. Moreover, clarity is needed because of multiple aspects and caveats of AI-generated questions, e.g:
 - is the question correct? do we have a score on this?
 - is the question culturally relevant at all?
 - is the generated language correct? especially in dialectal Arabic and low-resource languages, we need confirmation that LLMs can indeed write the question naturally.
 - is the QA according to the expectation, i.e., requires the image to be answered (see previous point).

While the human agreement is high, it would not be very informative if the definition of what is being measured is unclear.

**Questions:**

- Missing important details, e.g., l118 → how many were filtered or removed? I assume it is not removed since you want to keep the size of 310; in that case, how many were replaced?
- l130 → “which results in approximately 10 to 86 queries per topic.” The variance is extremely high; more discussion or details would be helpful (why is this the case, which dialect is affected, and is this due to LLM weakness on the task or something else?). Subsequently, how trustworthy is this process given such high variance?

Missing citation:
- In related works, datasets were discussed but not cited: Maya and PALO.
- Some other multilingual efforts for multimodal, cultural VQA: CVQA, SeaVQA.

---

> ### Author Response · Authors · 2025-11-20
> **Response to Y1YF: Rebuttal and Revision Summary (C1 to C2)**
>
> ### **Comment 1:**
> I’m concerned about the direction of this data. If the aim is to have a multimodal QA mainly focusing on visual, then we should expect the model to get ~0 performance if the question is only given in text. i.e., there is no image so you can’t really answer the question (as mentioned in l36 and the example in Figure 1)
>
> But, we see non-zero text-only performance, which means some questions are solvable without the image anyway, hence making this data not necessarily VQA. If the image is supposed to be an additional modality that can be utilized but not mandatory for answering, then this should be stated more explicitly without contradicting the examples (Figure 1)...
>
>
> ## **Response 1**
> Our goal is vision‑grounded QA, i.e., the image provides instance‑specific evidence; it is not a setting where language alone is always uninformative. While our benchmark is vision‑grounded, expecting ~0 accuracy from text‑only models is inconsistent with modern VQA practice [1, 2]: language priors and parametric knowledge enable non‑zero question‑only performance; accordingly, we report text‑only and text + image combinations and show substantial gains when the image is available, evidencing true visual grounding.
> In addition, the T/F and MCQ format gives the models some leverage to make educated guesses even with text only questions.
>
> Goyal et al. (2017) show that natural questions contain statistical cues ("Do you see…", "What sport…", etc.) that correlate with answers; models can exploit these even without seeing the image. This has been repeatedly documented. For example, in the seminal “balanced VQA” work, a language-only baseline achieved ~41–48% depending on the split [1].
>
> Schwenk et al. (2022) show that LLMs can answer many everyday visual questions from background knowledge (*"What color is a stop sign?"*, *"Which sport uses a bat?"*) without the particular image. On A-OKVQA, a question-only GPT-3 baseline reached ~35% on multiple-choice and ~11–13% on direct-answer—without images [2]
>
> Our contribution lies in proposing the **EverydayMMQA framework** and **OASIS dataset**. The framework facilitates the development of large-scale datasets like OASIS. It is one of the first large-scale tri-modal datasets containing aligned text–image–speech triplets and supporting multiple input-modality combinations, including text, image+text, speech, and speech+image. It comprises **0.92M images from 18 countries, 14.8K QA pairs in multiple language varieties, 20K synthetic audio samples for training, 263 hours of human-recorded audio for evaluation, and 105K QA pairs manually verified by annotators**. The proposed OASIS dataset is not only to benchmark the current VLM/MLLMs but also to train/fine-tune such models.
>
> [1] Goyal et al. "Making the v in vqa matter: Elevating the role of image understanding in visual question answering." CVPR. 2017.
> [2] Schwenk et al. "A-okvqa: A benchmark for visual question answering using world knowledge." In CCV, 2022.
>
> ---
> ### **Comment 2:** The implication on the improved result for T+I modality perhaps is not because of the “language burden” as stated. The boost may simply be because some questions require the image to be answered....
>
> ### **Response 2**
> We agree that the T+I gain comes from questions that require visual evidence (as illustrated in Fig. 1). Our definition of language burden is complementary: Arabic is more complex due to its rich morphology, diacritics, and orthographic variation, which makes text-only processing more challenging than in English. Based on our analysis, we observe that T+I performance for Arabic varieties is relatively higher than for English. To clarify this, we compute the absolute T→T+I differences across models (GPT-4.1 and GPT-5) for each language variety, as shown below. Across models, the T→T+I gain is consistently larger for Arabic (MSA, Egyptian, Levantine) than for English in accuracy-style tasks (e.g., MCQ and TF2), indicating that visual grounding helps mitigate Arabic-specific linguistic challenges.
>
> GPT-4.1 — Absolute gains (Δ)
>
> | Metric                    | English  | MSA     | Egyptian (ARZ) | Levantine (AJP) |
> | ------------------------- | -------- | ------- | --------------- | ---------------- |
> | **MCQ (Acc)**            | **+0.16** | **+0.22** | **+0.21** | **+0.21** |
> | **TF2 (Acc)**            | **+0.22** | **+0.32** | **+0.41** | **+0.43** |
>
>
> GPT-5 — Absolute gains (Δ)
>
> | Metric                    | English  | MSA     | Egyptian (ARZ) | Levantine (AJP) |
> | ------------------------- | -------- | ------- | --------------- | ---------------- |
> | **MCQ (Acc)**            | **+0.18** | **+0.23** | **+0.22** | **+0.22** |
> | **TF2 (Acc)**            | **+0.15** | **+0.42** | **+0.54** | **+0.54** |

---

> ### Author Response · Authors · 2025-11-20
> **Response to Y1YF: Rebuttal and Revision Summary (C3 to C4)**
>
> ### **Comment 3:**
> Another concern is on the LLM-generated data for benchmarking. The data is validated somehow (i.e., Likert scale), but it is not detailed enough, especially for machine-generated data. The Likert scale is not clear on what it measures (question difficulty? question quality? what is the definition of “quality”?). Providing the annotation guideline, including the exact Likert scale used, would be useful. Moreover, clarity is needed because of multiple aspects and caveats of AI-generated questions, e.g:
> * is the question correct? do we have a score on this?
> * is the question culturally relevant at all?
> * is the generated language correct? especially in dialectal Arabic and low-resource languages, we need confirmation that LLMs can indeed write the question naturally.
> * is the QA according to the expectation, i.e., requires the image to be answered (see previous point).
>
> While the human agreement is high, it would not be very informative if the definition of what is being measured is unclear.
>
> ### **Response 3**
> We have added an annotation guideline in Appendix G.
> * We have reported human annotation scores and agreement in Table 3. The question quality scores are higher >4.1 out of 5 for open-ended and MCQs.
> * For cultural relevance, we first generated topics grounded in cultural aspects, which are then manually verified. Based on the manually verified topics, we generated location-grounded queries. In each query, we then assigned a culturally relevant score (1-100) using GPT-4o. Based on our manual verification, we filtered queries that are lower than 80.
> * From English to dialectal translation, we provide results in Table 10, which suggest that GPT-4.1 is a reasonable choice for dialectal translation compared to English to MSA and dialectal translation.
> * As mentioned earlier, our focus was on vision-grounded QA. Current LLMs can answer a part of them based on language priors and parametric knowledge.
>
> ---
>
> ## **Comment 4** Missing important details, e.g., l118 → how many were filtered or removed? I assume it is not removed since you want to keep the size of 310; in that case, how many were replaced?
>
> ## **Response 4**
> In total, we have removed 135 topics from the complete set, as presented in Table 3 in the Appendix. We have not replaced any; we just removed them. Please see the country-wise stats below.
>
> | Country        | # of Topics | # of Topics after manual verification | Removed |
> |----------------|-------------|----------------------------------------|---------|
> | Algeria        | 310         | 304                                    | 6       |
> | Bahrain        | 310         | 305                                    | 5       |
> | Egypt          | 310         | 310                                    | 0       |
> | Iraq           | 310         | 303                                    | 7       |
> | Jordan         | 310         | 297                                    | 13      |
> | Kuwait         | 310         | 301                                    | 9       |
> | Lebanon        | 310         | 306                                    | 4       |
> | Libya          | 310         | 307                                    | 3       |
> | Morocco        | 310         | 310                                    | 0       |
> | Oman           | 310         | 309                                    | 1       |
> | Palestine      | 310         | 294                                    | 16      |
> | Qatar          | 310         | 310                                    | 0       |
> | Saudi Arabia   | 310         | 287                                    | 23      |
> | Sudan          | 310         | 310                                    | 0       |
> | Syria          | 310         | 283                                    | 27      |
> | Tunisia        | 310         | 303                                    | 7       |
> | UAE            | 310         | 310                                    | 0       |
> | Yemen          | 310         | 296                                    | 14      |
> | **Total**      | **5,580**   | **5,445**                               | **135** |

---

> > ### Author Response · Authors · 2025-11-20
> > **Response to Y1YF: Rebuttal and Revision Summary (C5 to C6)**
> >
> > ## **Comment 5** l130 → “which results in approximately 10 to 86 queries per topic.” The variance is extremely high; more discussion or details would be helpful (why is this the case, which dialect is affected, and is this due to LLM weakness on the task or something else?).
> >
> > ## **Response 5**
> > Thank you for pointing out the issue with the number of queries; the previous values were incorrect. We have revised Section 2 and updated Table 3 (Section A.1 in the appendix) to include the minimum, maximum, and standard deviation of country-wise queries. We also added two new figures (Figures 4 and 5) showing category-wise coverage. In addition, we have revised Section 2.1 and provided a more detailed clarification.
> >
> > On average across countries, the mean ± std is 175.1 ± 14.5. Across categories, it is 174.8 ± 13.18. The variance in the number of queries is a result of the image-search–based filtering process. This behaviour is expected; some subcategories, such as *Famous Landmarks*, *Heritage & History*, are highly visual and associated with many distinct country-specific entities that yield diverse and relevant images. In contrast, some subcategories are less naturally grounded in images; for example, *eSports & Gaming* produced the fewest queries due to its lower visual relevance. This variation also reflects the underrepresentation of digital visual content for some countries
> >
> > ***Subsequently, how trustworthy is this process given such high variance?***
> >
> > As responded in *comment 2 of Reviewer Bqyn*, in the *"Query Generation and Filtering"*, we perform below steps:
> >
> > 1. We initially use GPT-4o to generate topics with the instruction to “ensure that the topics reflect the cultural, historical, or modern significance of the specified location.” These topics are manually revised and validated by knowledgeable contributors from each of the 18 countries, who remove inappropriate or generic topics. Only these human-verified topics are used downstream.
> > 2. From the validated topics, we generate queries and then use GPT-4o to assign relevancy scores. A sample of queries and their scores was manually inspected, and we empirically selected a conservative threshold of 80/100 to discard low-relevance queries.
> >
> > This approach has been validated for 18 countries, which ensures higher reliability of the process.
> >
> > ---
> > ## **Comment 6:**
> > Missing citation:
> > * In related works, datasets were discussed but not cited: Maya and PALO.
> > * Some other multilingual efforts for multimodal, cultural VQA: CVQA, SeaVQA.
> >
> > ## **Response 6**
> > We have updated the related work section and included detailed citations for the referenced studies in Table 15.

---

> ### Author Response · Authors · 2025-11-24
> **Responses to Comments (Y1YF) – follow up**
>
> Dear Reviewer Y1YF,
>
> Thank you for your valuable feedback and time. We would like to request that if you have any other concerns please let us know. We would be happy to clarify.
>
> Thank you
>
>
> Best Regards
>
> The Authors

---

> > ### Comment · Reviewer_Y1YF · 2025-11-24
> >
> > The additional experiments and results are very helpful. Thank you for including them.
> >
> > However, my concern about T+I remains. I am still not convinced that VQA data which can be answered without the image constitutes good VQA data. In fact, the statement from Goyal (2017) should be addressed as highlighting a flaw in some of VQA dataset, an artifact that models can exploit. Similarly, Schwenk (2022) expresses the same view. Table 4 in Schwenk (2022) shows examples of data they reject, including a case of "Question doesn’t require looking at the image to answer.". This indicates that VQA questions must rely on both modalities to be considered good data.

---

> > > ### Author Response · Authors · 2025-11-26
> > > **Responses to the follow-up comments (Y1YF)**
> > >
> > > ## **Responses:**
> > >
> > > We thank the reviewer for the follow-up and fully share the concern about language priors in VQA. We agree that high-quality VQA / vision-grounded QA data should be visually grounded. However, we respectfully note that **requiring zero text-only accuracy is neither necessary nor realistic** for defining good VQA questions.
> > >
> > > ### **Principled reasoning:**
> > >
> > > A core principle of VQA is that the image should provide instance-specific evidence that substantially improves performance, which can be measured by comparing text-only and text+image accuracy. Non-zero text-only accuracy reflects language priors or background knowledge and does not invalidate the dataset. In fact, **measuring the text-only baseline is essential for understanding model reliance on priors.**
> > >
> > > ### **VQA datasets in practice:**
> > >
> > > **VQA v1:** Antol et al. [1] report that humans answer ~41% of questions correctly without seeing the image versus ~83% with the image. This non-zero question-only accuracy, together with large gains from adding the image, is treated as evidence of visual grounding rather than a flaw.
> > >
> > > **VQA v2 (Goyal et al., 2017) [2]:** Designed to mitigate priors, but question-only models still achieve ~43%, while full models reach ~52–59%. Language-only baselines are explicitly used to measure priors, not to define *"good"* VQA questions.
> > >
> > > **A-OKVQA (Schwenk et al., 2022) [3]:** Even with instructions to require image observation, question-only baselines (BERT, CLIP-text, GPT-3) reach ~33–39%, whereas text+image models exceed 50% (Table 3).
> > >
> > > ### **Our benchmark philosophy:**
> > >
> > > Following this established practice, we:
> > >
> > > - Write questions about the image and evaluate them for visual relevance.
> > > - Accept that large language models or humans may answer some questions using priors.
> > > - Measure grounding by the gain from adding the image over a text-only baseline.
> > >
> > > This approach ensures that the benchmark encourages models to integrate both modalities, rewards vision grounding, and quantifies reliance on language priors. Eliminating all questions answerable from text alone would reduce dataset diversity and real-world relevance.
> > >
> > > ### **Empirical illustration:**
> > >
> > > We provided examples, in the link below, where:
> > > - A language model can guess an answer from text alone (Example 1).
> > > - The same model fails without the image, showing the need for visual evidence (Example 2).
> > >
> > > [https://limewire.com/d/2EDM3#pmHIz074p8](https://limewire.com/d/2EDM3#pmHIz074p8)
> > >
> > > ***In summary***, our dataset aligns with current VQA/vision-grounded QA best practices: it requires visual grounding at the dataset level, explicitly measures language priors, and rewards models that integrate both modalities. Non-zero text-only performance does not contradict this goal. It is an expected and informative aspect of high-quality VQA datasets.
> > >
> > > - [1] Antol et al. "VQA: Visual question answering." In Proceedings of the IEEE international conference on computer vision, pp. 2425-2433. 2015.
> > > - [2] Goyal et al. "Making the v in vqa matter: Elevating the role of image understanding in visual question answering." In CVPR, pp. 6904-6913. 2017.
> > > - [3] Schwenk et al., "A-okvqa: A benchmark for visual question answering using world knowledge." In ECCV, pp. 146-162. Cham: Springer Nature Switzerland, 2022.

---

### Official Review · Reviewer_Bqyn · 2025-11-01

**Soundness:** 2
**Presentation:** 3
**Contribution:** 2
**Rating:** 4
**Confidence:** 3

**Summary:**

The paper proposes EveryDayMMQA, a framework for building culturally grounded multimodal question-answering datasets, which consists of (i) image collection/filtering (ii) metadata and QA generation, (iii) speech generation/recording (iv) translation and (v) quality assurance. The authors use the proposed framework (EveryDayMMQA) to build OASIS, which is a large multimodal dataset that integrates speech, images and text, focused on english and Arabic, covering 18 countries. The proposed dataset supports four inputs: text, speech, text+image and speech+image, covering different type of questions, Open-ended, Multiple choice and True/False. With the proposed dataset the authors perform a variety of experiments with open and closed source models, additionally they finetune one model and show that this achieves a higher accuracy on cultural knowledge, specially when the question is image-grounded.

**Strengths:**

Paper Strenghts:
1. The paper extends the common modalities of image-text in cultural-related resources for VQA, and add speech, targeting Arabic dialects alongside English.
2. With the proposed pipeline the authors are able to build Oasis, a large scale multimodal dataset providing a testbed for multimodal and culturally diverse evaluation. EveryDayMMQA helps for gathering culturally related data, which is usually difficult, specially for low-resource languages.
3. Broad evaluations with open and closed-source models.

**Weaknesses:**

Paper Weaknesses:
1. Section 2 would benefit for a brief general description of the method before describing it in detail, otherwise is just a bit complicated at first to understand why you are doing each part. On the other hand, when each part is described like in "Topic Generation and Filtering" or "Query Generation and Filtering", please refer where the reader can see those things that you are describing, like in what part of the paper the reader can see those topics or queries, this makes it easier to understand.
2. In Section 2 "Query Generation and Filtering" the authors say that they use "GPT-4o to assign a cultural-relevancy score, which reflects the cultural fit". Page 19 shows the prompt used for this, which contains: "Ensure that the topics reflect the cultural, historical or modern significance of the specified location". It's well know that VLMs lack cultural knowledge even the biggest closed source models from the GPT family, multiple works benchmark them in cultural-related VQA, for example [1] CVQA, so using them to give a cultural-relevancy score seems to not be really appropriate.
3. Section 2.2 describes the process to get images via country-localized search using Google Custom Search, in my personal experience getting images related to culture for low-resource cultures is hard, which can be the case of Arabic countries, which is why some works get images directly from annotators like in [1]. Perhaps there were difficulties related to this?
4. More examples of dataset questions would benefit the paper.
5. Section 2.6 describes translation from English into MSA, it is also known [1] that using translation fails to capture cultural nuances inherent in different languages, was this considered ?.

[1] https://arxiv.org/pdf/2406.05967

**Questions:**

Please refer the weaknesses for questions and doubts. Overall, I would like to see the authors responses in using VLLMs/LLMs to do a dataset related to culture, when is well known that these models still lack cultural knowledge. Sorry if I misunderstood something, currently I'm giving borderline reject, but after rebuttal I will revise my decision.

---

> ### Author Response · Authors · 2025-11-20
> **Response to Bqyn: Rebuttal and Revision Summary (C1 to C4)**
>
> ## **Comment 1:** Section 2 would benefit for a brief general description of the method before describing it in detail, otherwise is just a bit complicated at first to understand why you are doing each part. On the other hand, when each part is described like in "Topic Generation and Filtering" or "Query Generation and Filtering", please refer where the reader can see those things that you are describing, like in what part of the paper the reader can see those topics or queries, this makes it easier to understand.
>
> ---
> ## **Response 1**
> We have revised Section 2. Each module of the EverydayMMQA framework (Figure 2) is now aligned with the subsection titles to make the section easier to read. Paragraph titles are matched with the corresponding parts of Figure 2. For example, Section 2.1 is mapped to the first component of Figure 2, *Topic & Image Query Generation*. We have also revised Figure 2 for enhanced clarity.
>
> ---
> ## **Comment 2: ** In Section 2 "Query Generation and Filtering" the authors say that they use "GPT-4o to assign a cultural-relevancy score, which reflects the cultural fit". Page 19 shows the prompt used for this, which contains: "Ensure that the topics reflect the cultural, historical or modern significance of the specified location". It's well know that VLMs lack cultural knowledge even the biggest closed source models from the GPT family, multiple works benchmark them in cultural-related VQA, for example [1] CVQA, so using them to give a cultural-relevancy score seems to not be really appropriate.
>
> ---
>
> ## **Response 2**
> We fully agree that current VLMs, including GPT-4o models, have limitations in cultural knowledge, and that relying on them alone for *"cultural relevance"* would be problematic. In our pipeline, however, GPT-4o is only used to facilitate the generation and filtering mechanisms.
>
> Concretely, in the *Query Generation and Filtering*, we followed the steps below:
>
> 1. We initially use GPT-4o to generate topics with the instruction to “ensure that the topics reflect the cultural, historical, or modern significance of the specified location.” These topics are manually revised and validated by knowledgeable contributors from each of the 18 countries, who remove inappropriate or generic topics. Only these human-verified topics are used downstream.
>
> 2. From the validated topics, we generate queries and then use GPT-4o to assign relevancy scores. A sample of queries and their scores was manually inspected, and we empirically selected a conservative threshold of 80/100 to discard low-relevance queries.
>
> We also note that, despite their limitations, GPT-4o models currently achieve state-of-the-art performance on cultural VQA benchmarks such as CVQA (e.g., 75.4 for English and 74.3 for location-specific prompts, as reported in Table 3 of the CVQA paper). For our work, *human-in-the-loop* and very conservative *filtering* mechanisms ensure higher quality of the data.
>
> ---
> ## **Comment 3:** Section 2.2 describes the process to get images via country-localized search using Google Custom Search, in my personal experience getting images related to culture for low-resource cultures is hard, which can be the case of Arabic countries, which is why some works get images directly from annotators like in [1]. Perhaps there were difficulties related to this?
>
> ---
> ## **Response 3**
> Indeed, obtaining culturally rich images with web search can be challenging, especially for low-resource languages; however, it also depends on the highly relevant image-based queries. We developed an iterative, *human-in-the-loop* process. For the first country we worked on, we went through several rounds of refinement at each step, such as category and sub-category selection, topic generation, and finally query generation, together with native contributors/annotators. Once this approach was stabilised, we scaled it to the remaining countries, again with manual checks and adjustments where needed.
>
> We fully agree that directly collecting images from annotators, as done in CVQA [1], can produce highly authentic, culturally grounded content. However, this approach is difficult to scale to the size we target (e.g., CVQA contains 5,239 images and 10,374 questions), whereas our Google Custom Search–based approach, combined with iterative query refinement and human review, allows us to reach a much larger coverage across 18 countries.
>
> ---
>
> ## **Comment 4:** More examples of dataset questions would benefit the paper.
>
> ## **Response 4**
> Due to space limitations, we provided one example in *Figure 1* in the main paper. We added more examples in *Appendix H*.

---

> > ### Author Response · Authors · 2025-11-20
> > **Response to Bqyn: Rebuttal and Revision Summary (C5 to C6)**
> >
> > ## **Comment 5:** Section 2.6 describes translation from English into MSA, it is also known [1] that using translation fails to capture cultural nuances inherent in different languages, was this considered ?.
> > [1] https://arxiv.org/pdf/2406.05967
> >
> > ## **Response 5**
> > For English-to-MSA translation, we used an in-house LLM-based system whose performance was evaluated on 11 datasets, achieving an average BLEU score of 25.11 compared to 19.62 for Google Translate. For dialectal translation, we used GPT-4.1; the results reported in Table 8 show its reasonable performance on dialectal translation compared to English -> MSA -> dialects (Egy, Lev). For MT, we used state-of-the-art models and incorporated human-in-the-loop setups to ensure higher quality.
> >
> > ---
> >
> > ## **Comment 6:** Please refer the weaknesses for questions and doubts. Overall, I would like to see the authors responses in using VLLMs/LLMs to do a dataset related to culture, when is well known that these models still lack cultural knowledge. Sorry if I misunderstood something, currently I'm giving borderline reject, but after rebuttal I will revise my decision.
> >
> > ## **Response 6**
> > Thank you for your consideration. We have addressed all of your concerns. We kindly ask you to review the responses, and if you have any further questions, please let us know. We will be happy to clarify.

---

> ### Author Response · Authors · 2025-11-24
> **Responses to Comments (Bqyn) – follow up**
>
> Dear Reviewer Bqyn,
>
> Thank you for your valuable feedback and time. We would like to request that if you have any other concerns please let us know. We would be happy to clarify.
>
> Thank you
>
> Best Regards
>
> The Authors

---

### Official Review · Reviewer_t9AU · 2025-11-01

**Soundness:** 4
**Presentation:** 2
**Contribution:** 2
**Rating:** 4
**Confidence:** 5

**Summary:**

The paper proposes "EverydayMMQA," a framework for generating large-scale, multimodal, multilingual question-answering datasets. Using this framework, the authors create "OASIS," a dataset with approximately 0.92 million images and 14.8 million QA pairs focused on English and several Arabic varieties, covering 18 Arab countries. A main feature of OASIS is the inclusion of spoken questions (both synthetic and human-recorded), enabling evaluation across four input modalities (speech-only, text-only, speech+image, text+image). The authors benchmark several closed and open-source models on OASIS and conduct a fine-tuning experiment. Their main findings suggest that visual grounding is the most critical factor for improving performance and that fine-tuning on their dataset enhances a model's cultural awareness.

**Strengths:**

- The paper's goal is certainly well-motivated. The lack of culturally diverse, spoken-language multimodal benchmarks is a clear gap.

- The development of the EverydayMMQA framework is a valuable contribution on its own. By modularizing the process into distinct stages (e.g., query generation, image retrieval, QA generation, quality control), the authors provide a scalable and reusable methodology that other researchers can adopt for creating similar resources for different cultural or linguistic contexts.

- The inclusion of spoken questions, both synthetic and human-recorded, is a major differentiator from most existing VQA datasets. This enables a much more comprehensive evaluation of "omni" models that are designed to handle multiple input modalities simultaneously.

- In experiments, authors test models across four different input combinations, which allows for a clear analysis of the impact of each modality. The conclusion that visual grounding acts as an "equalizer" that mitigates the performance loss from speech input and ASR errors is a particularly strong and interesting result. The fine-tuning experiment also effectively demonstrates the utility of the dataset for improving model performance.

**Weaknesses:**

- The framwork of the data construction appears to be a description of a specific, multi-stage data collection pipeline that relies heavily on a series of LLM API calls. It is not clear what makes this a generalizable or reusable "framework" beyond the specific implementation for this paper. The contribution seems to be the dataset itself, and framing the pipeline as a separate, novel framework feels like an overstatement.

- A major concern is the almost complete reliance on automated generation for all critical components of the dataset. The cultural topics, search queries, image descriptions, and the QA pairs themselves are all generated by large language models. The paper argues it is addressing the "Western-centric bias" of existing resources, but it does so by using the very same large, predominantly Western-trained models to generate its "culturally grounded" content. It is highly questionable whether a model like GPT-4 can genuinely produce authentic cultural nuances for 18 different Arab countries. This methodology risks creating a large dataset of plausible-sounding but ultimately synthetic and stereotyped content, rather than a truly culturally grounded resource. The "quality check" also relies on LLMs, which compounds this issue.

- The vast majority of the spoken data (~20,000 hours) is synthetic, generated by a TTS model. The paper’s own analysis (Appendix A.3) shows that this synthetic speech is much "cleaner" and easier for an ASR model to process than the small amount of human-recorded speech. This means the benchmark is not truly evaluating performance on real-world "spoken" QA, but rather on "TTS-based" QA. The conclusions drawn from the speech modality are therefore of limited applicability to realistic, noisy human-computer interaction scenarios.

- The main conclusions from the experiments are not particularly surprising. It is expected that providing an image for a visual question (the T+I condition) would drastically improve performance over text-alone (the T condition). This finding, while true, is largely confirmatory. More concerning is the fine-tuning experiment. The authors fine-tune a model on a subset of the automatically generated training data and show that its performance improves on the automatically generated test set. This result is at high risk of being circular. It may primarily demonstrate that the model has learned to replicate the specific patterns, style, and potential artifacts of the data generator, rather than acquiring genuine cultural or multimodal reasoning abilities.

- Furthermore, this paper does not cite two very relevant papers, authors must explain why:

Romero, David, et al. "CVQA: culturally-diverse multilingual visual question answering benchmark." Proceedings of the 38th International Conference on Neural Information Processing Systems. 2024.

Vayani, Ashmal, et al. "All languages matter: Evaluating lmms on culturally diverse 100 languages." Proceedings of the Computer Vision and Pattern Recognition Conference. 2025.

**Questions:**

N/A

---

> ### Author Response · Authors · 2025-11-20
> **Response to t9AU: Rebuttal and Revision Summary (C1 to C2)**
>
> ## **Comment 1:** The framwork of the data construction appears to be a description of a specific, multi-stage data collection pipeline that relies heavily on a series of LLM API calls. It is not clear what makes this a generalizable or reusable "framework" beyond the specific implementation for this paper. The contribution seems to be the dataset itself, and framing the pipeline as a separate, novel framework feels like an overstatement.
>
> ---
>
> ## **Response 1**
> The ***framework*** is language and location-agnostic. It is a modular, configurable pipeline rather than a single hard-coded implementation: users can plug in different locations, languages, category taxonomies, and LLM backends without changing the overall structure.
>
> It has been tested for 18 different countries, and can be reused for other regions and languages by: (i) specifying the target location, (ii) defining or importing category/subcategory schemas, and (iii) setting the working language(s) in the prompts and filtering modules.
> The modules for category, topic generation, query generation, location-grounded image collection, and quality filtering are all parameterizable rather than dataset-specific. LLMs are used as interchangeable components for generation and filtering, with humans in the loop at key stages for quality control. For a large-scale multimodal location and culturally grounded dataset curation, the framework is very seamless.
>
> ---
> ---
> ## **Comment 2:** A major concern is the almost complete reliance on automated generation for all critical components of the dataset. The cultural topics, search queries, image descriptions, and the QA pairs themselves are all generated by large language models. The paper argues it is addressing the "Western-centric bias" of existing resources, but it does so by using the very same large, predominantly Western-trained models to generate its "culturally grounded" content. It is highly questionable whether a model like GPT-4 can genuinely produce authentic cultural nuances for 18 different Arab countries. This methodology risks creating a large dataset of plausible-sounding but ultimately synthetic and stereotyped content, rather than a truly culturally grounded resource. The "quality check" also relies on LLMs, which compounds this issue.
>
> ---
> ## **Response 2**
> In our work, GPT-based models are primarily used as tools to assist data filtering and curation, not as the sole source to develop the whole QA dataset. We explicitly design multiple human-in-the-loop stages to ensure location- and culture-grounded content. Concretely:
>
> - (i) Initial topic candidates are manually revised and validated by knowledgeable contributors from each of the 18 countries, who remove inappropriate or generic topics and add locally salient ones. For example, the topic, “Berber Village of Aït Benhaddou”, is removed from the Algeria collection as the location is not relevant to Algeria.
>
> - (ii) Queries are then generated conditioned on these human-verified topics and are explicitly location- and culture-aware. We apply an automatic relevance scoring step using GPT-4o and adopt a conservative threshold (80/100), which we selected based on manual checking to balance coverage and specificity.
>
> - (iii) Location-specific images are enforced by using geo-parameterised Google image search tied to the target country, so that the visual content is grounded in that locale rather than being globally generic.
>
> For QA quality, we combine human and LLM-based assessment, as outlined in Section 2.8.
>
> - (i) In the human annotation study, we annotated 13,728 QA pairs across 13 countries for question and answer clarity/quality using a Likert scale (guidelines in Appendix G). Each QA was annotated by two human annotators. As reported in Lines 267-271, inter-rater agreement is relatively higher (e.g., both question and answer quality have r_wg > 0.86), suggesting that the QAs are generally judged as clear and acceptable by humans.
>
> - (ii) We then use LLM-based annotation (Gemini-2.5-Pro and Llama-4-Scout 17B-16E expert) to scale up the evaluation to the full test set (139,720 QA pairs associated with 34,930 image samples – test set), not to replace human judgment but to extend it. We also measure LLM-human agreement: the correlations for question quality, answer quality, rationale plausibility & faithfulness, and rationale clarity & informativeness are 0.93, 0.87, 0.86, and 0.93, respectively, indicating that LLM scores are reasonably aligned with human ratings.

---

> > ### Author Response · Authors · 2025-11-20
> > **Response to t9AU: Rebuttal and Revision Summary (C3 to C5)**
> >
> > ## **Comment 3:** The vast majority of the spoken data (~20,000 hours) is synthetic, generated by a TTS model. The paper’s own analysis (Appendix A.3) shows that this synthetic speech is much "cleaner" and easier for an ASR model to process than the small amount of human-recorded speech. This means the benchmark is not truly evaluating performance on real-world "spoken" QA, but rather on "TTS-based" QA. The conclusions drawn from the speech modality are therefore of limited applicability to realistic, noisy human-computer interaction scenarios.
> >
> > ---
> > ## **Response 3**
> > Our core spoken QA benchmark and results are computed on human-recorded speech consisting of 141 hours of speech. *The large-scale synthetic speech (~20,000 hours) is primarily intended for training.*
> >
> > Specifically, all results we report for the spoken modality, such as speech (S), transcription (Tr), speech + image (S+I), and transcription + image (Tr+I), in Figure 3, Table 12, and Figure 6 are based on human recordings in the test set, not on TTS audio. Thus, *the benchmarked performance we discuss in the main paper reflects models’ behaviour on real, **human-produced speech** rather than on cleaner TTS signals.*
> >
> > We have evaluated human recordings using a publicly accessible ASR system released as a part of the Fanar platform (https://fanar.qa/)  for Arabic and widely used whisper model from openAI. We have updated the results in Appendix A.4.
> >
> > ---
> > ## **Comment 4:** The main conclusions from the experiments are not particularly surprising. It is expected that providing an image for a visual question (the T+I condition) would drastically improve performance over text-alone (the T condition). This finding, while true, is largely confirmatory. More concerning is the fine-tuning experiment. The authors fine-tune a model on a subset of the automatically generated training data and show that its performance improves on the automatically generated test set. This result is at high risk of being circular. It may primarily demonstrate that the model has learned to replicate the specific patterns, style, and potential artifacts of the data generator, rather than acquiring genuine cultural or multimodal reasoning abilities.
> >
> > ---
> > ## **Response 4**
> > We agree that, conceptually, it is expected that providing an image for a genuinely visual question (T+I) should outperform text-only (T). Our aim is to quantify how much different models benefit from visual input across countries and question types in a culturally grounded setting; here, we are providing a benchmark.
> >
> > As for the dataset quality, we conduct extensive human evaluation of the QAs (Section 2.8): human annotators report high agreement and relatively high scores for question quality, answer quality, rationale plausibility & faithfulness, and rationale clarity & informativeness (0.93, 0.87, 0.86, and 0.93, respectively), indicating that the target labels are generally meaningful and not dominated by obvious artifacts.
> >
> > Regarding the *fine-tuning experiment*, our goal is not to argue that the model acquires deep cultural or multimodal reasoning solely from this setup, but rather to demonstrate that the dataset is learnable and can be used to improve performance on the tasks it defines. However, to assess generalizability and address this comment, we curated the Arabic portion of the ALM-Bench [1] dataset, which contains 477 images with question types similar to OASIS, such as open-ended, MCQ, and True/False. We benchmarked the Qwen-3B base model and our fine-tuned model with ALM-Bench for both the short and long open-ended question sets. The overall LLM-as-a-judge scores are 4.62 for the base model and 5.85 for our fine-tuned model, indicating improved capability when trained on the OASIS dataset.
> >
> > We also want to highlight that our use of LLMs for filtering and QA generation is aligned with recent work that adopts similar practices with GPT-4o models [1, 2, 3], while emphasising that human evaluation remains central to validating the quality of our dataset.
> >
> > 1. Vayani et al. "All languages matter: Evaluating lmms on culturally diverse 100 languages." CVPR. 2025.
> > 2. Yue et al. "Pangea: A fully open multilingual multimodal llm for 39 languages." ICLR, 2024.
> > 3. Urailertprasert et al. "Sea-vqa: Southeast asian cultural context dataset for visual question answering." In ALVR, 2024.
> > ---
> > ## **Comment 5:** Furthermore, this paper does not cite two very relevant papers, authors must explain why:
> > * Romero, David, et al. "CVQA: culturally-diverse multilingual visual question answering benchmark."  2024.
> > * Vayani, Ashmal, et al. "All languages matter: Evaluating lmms on culturally diverse 100 languages." . 2025.
> >
> > ---
> > ## **Response 5**
> >
> > We apologise for this unintentional mistake in not citing the CVQA paper. We have cited Vayani et al. in Table 15. In the revised version of the paper, we have included the CVQA paper in Table 15.

---

> ### Author Response · Authors · 2025-11-24
> **Responses to Comments (t9AU) – follow up**
>
> Dear Reviewer t9AU,
>
> Thank you for your valuable feedback and time. We would like to request that if you have any other concerns please let us know. We would be happy to clarify.
>
> Thank you
>
> Best Regards
>
> The Authors

---

### Official Review · Reviewer_rPig · 2025-11-01

**Soundness:** 3
**Presentation:** 3
**Contribution:** 3
**Rating:** 4
**Confidence:** 4

**Summary:**

The paper presents EverydayMMQA, a systematic and scalable framework for building multimodal, multilingual, and culturally grounded QA datasets. OASIS, the resulting resource, integrates 0.92M images, 14.8M QA pairs, and 3.7M spoken questions, spanning English, Modern Standard Arabic, Egyptian, and Levantine Arabic. Four unique input modes are supported: text-only, speech-only, text+image, and speech+image. The dataset is annotated for cultural nuance, with QA pairs grouped by knowledge vs. commonsense reasoning, semantic domain, and cognitive focus. Robust benchmarking of state-of-the-art closed (e.g., GPT-4.1, Gemini 2.5 Pro) and open (e.g., Qwen2.5, Phi-4) models demonstrates significant gains.

**Strengths:**

1. EverydayMMQA stands out through its tri-modal design (text, image, and speech) and explicit attention to diverse cultural contexts across the Arab world, including multiple dialects.
2. The framework involves LLM-driven topic/query/image filtering, human-in-the-loop validation, manual and LLM-based annotation, and multi-layered quality control.
3. Finetuning on a smaller subset of OASIS shows improvement in model performance, indicating its usability as a training dataset.

**Weaknesses:**

1. The authors rely on GPT-4o for assigning cultural relevance scores; however, they do not study how effective it is at this task. This overreliance on GPT-4o's capability in a non-English setting could be a problem.
2. The authors tested the finetuned model on the same data source (OASIS). The paper would benefit from an understanding of the model's generalizability, for instance, if a model trained on OASIS were tested on different benchmarks.
3. The analysis in Section 4 is good at dissecting performance by modality (T vs. S vs. T+I) and language (En vs. MSA). However, it stops short of a deeper qualitative analysis of the cultural dimension. Even with visual grounding (T+I), models are not perfect. The paper would benefit if it included examples of culturally nuanced questions that the best models still get wrong.

**Questions:**

1. Line 72 states there are 16M QAs, which contradicts the 14.8M QA pairs mentioned in the abstract. Please clarify this discrepancy.
2. Given the paper's focus on dialectal diversity, are there plans to expand the spoken dataset to include Arabic dialects?
3. Could you provide a qualitative error analysis for the best-performing models (e.g., GPT-4.1 with T+I)? What specific types of cultural, pragmatic, or commonsense reasoning do they still fail at, even with visual grounding?
4. Can you please clarify the dataset size? Is the 14.8M figure the total number of text QA pairs (i.e., 3.7M unique QA sets $\times$ 4 language varieties)?

---

> ### Author Response · Authors · 2025-11-20
> **Response to rPig: Rebuttal and Revision Summary (C1 to C5)**
>
> ### **Comment 1:** The authors rely on GPT-4o for assigning cultural relevance scores; however, they do not study how effective it is at this task. This overreliance on GPT-4o's capability in a non-English setting could be a problem.
>
> ---
> ### **Response 1**
> We agree that “overreliance on GPT-4o's capability” can be an issue. Therefore, to mitigate this, at every step of our ***EverydayMMQA framework***, we incorporated ***human-in-the-loop*** checks at multiple stages to ensure quality:
> - topics were manually verified and curated before query generation;
> - the cultural relevance threshold was calibrated using manual checks of scored queries;
> - the final benchmark used for evaluation is based on human validation.
>
> In particular, we conducted a *human evaluation* on a subset of the test sets from each country, comprising 110K annotations on 13,728 sample images. Details on annotation quality and agreement scores are reported in Sections 2.8 and A.3.
>
> Within the ***Topic & Image Query Generation module*** of our framework, we used GPT-4o in two ways:
> - to generate queries, where we explicitly instruct the model to ground its outputs in the target country, category, subcategory, and topic, and to incorporate localized and culturally relevant elements; and
> - to assign a cultural relevance score that is used only as a filtering mechanism to discard less relevant queries.
>
> After query generation, GPT-4o assigns a score in the range [0, 100]. Based on a manual inspection of a sample of queries across countries and categories, we set a threshold of ≥80/100. Applying this threshold reduced the pool from 110,126 to 97,678 queries.
>
> ---
>
> ### **Comment 2:** The authors tested the finetuned model on the same data source (OASIS). The paper would benefit from an understanding of the model's generalizability, for instance, if a model trained on OASIS were tested on different benchmarks.
>
> ---
>
> ### **Response 2**
> To assess generalizability and address this comment, we curated the Arabic portion of the ALM-Bench [1] dataset, which contains 477 images with question types similar to OASIS, such as open-ended, MCQ, and True/False. We benchmarked the Qwen-3B base model and our fine-tuned model with ALM-Bench for both the short and long open-ended question sets. The overall LLM-as-a-judge scores are 4.62 for the base model and 5.85 for our fine-tuned model, indicating improved capability when trained on the OASIS dataset.
>
> 1. Vayani et al. "All languages matter: Evaluating lmms on culturally diverse 100 languages.", CVPR, 2025.
>
> ---
> ### **Comment 3:** The analysis in Section 4 is good at dissecting performance by modality (T vs. S vs. T+I) and language (En vs. MSA). However, it stops short of a deeper qualitative analysis of the cultural dimension. Even with visual grounding (T+I), models are not perfect. The paper would benefit if it included examples of culturally nuanced questions that the best models still get wrong.
>
> ---
>
> ### **Response 3**
> We conducted an error analysis to understand the types of images on which the models failed. In Section E of the Appendix, Figure 9 presents examples of images where almost all models were unable to provide correct answers. These cases suggest that the models require additional contextual information to handle such cultural nuances. We also provide examples in Figure 10 where the fine-tuned model successfully answered the questions.
>
>
> ---
>
> ### **Comment 4:** Line 72 states there are 16M QAs, which contradicts the 14.8M QA pairs mentioned in the abstract. Please clarify this discrepancy.
>
> ### **Response 4**
> Thanks for the pointer. This is a typo; we revised it in the paper. Correct number is 14.8M QA pairs.
>
> ---
>
> ### **Comment 5:** Given the paper's focus on dialectal diversity, are there plans to expand the spoken dataset to include Arabic dialects?
>
> ### **Response 5**
> In the current version of the paper, we already include Egyptian and Levantine dialects for QA. We are also working on dialectal support for synthetic voice generation and will incorporate these in future studies. For dialectal voice recording, we are currently focusing on Egyptian, and we will report these results in the updated version.
>
> ---

---

> > ### Author Response · Authors · 2025-11-20
> > **Response to rPig: Rebuttal and Revision Summary (C6 to C7)**
> >
> > ### **Comment 6:** Could you provide a qualitative error analysis for the best-performing models (e.g., GPT-4.1 with T+I)?  What specific types of cultural, pragmatic, or commonsense reasoning do they still fail at, even with visual grounding?
> >
> > ### **Response 6**
> > We added our analysis in Section E, presenting category- and subcategory-wise results. Our findings indicate that models perform well in certain categories (e.g., *Heritage & History*) but struggle in others (e.g., *eSports & Gaming*), as shown in Figure 11. We also present a category-wise comparison demonstrating that the fine-tuned model substantially improves performance in the *Objects* category, as shown in Figure 12. This further confirms the importance of domain-specific model training.
> >
> > ---
> > ### **Comment 7:** Can you please clarify the dataset size? Is the 14.8M figure the total number of text QA pairs (i.e., 3.7M unique QA sets × 4 language varieties)?
> >
> > ### **Response 7**
> > Yes. The 14.8M QA pairs correspond to four language varieties, derived from approximately 3.7M unique QA sets. We have updated the figure illustrating the OASIS dataset overview accordingly.

---

> ### Author Response · Authors · 2025-11-24
> **Responses to Comments – follow up**
>
> Dear Reviewer rPig,
>
> Thank you for your valuable feedback and time. We would like to request that if you have any other concerns please let us know. We would be happy to clarify.
>
> Thank you
>
> Best Regards
>
> The Authors

---

### Author Response · Authors · 2025-11-20
**Summary of Revisions and Responses to All Reviewers, AC, and SAC**

Dear Reviewers, AC, and SAC,

We appreciate the time and effort you dedicated to evaluating our submission. We have carefully addressed all concerns raised in the individual reviews, and we have updated the manuscript accordingly. All revised or newly added text is highlighted in **blue** in the updated version.

Below, we summarise the key contributions of our work, which we believe collectively strengthen the novelty, importance, and potential impact of this research:

* **A language-agnostic, scalable framework (EverydayMMQA)** for curating tri-modal data (text, image, speech) from **any geography and any language**. The framework’s scalability and extensibility distinguish it from prior multimodal data collection efforts.

* **One of the first large-scale tri-modal datasets, OASIS**, containing aligned text–image–speech triplets and supporting multiple input-modality combinations, including **text**, **image+text**, **speech**, and **speech+image**. It comprises **0.92M images from 18 countries**, **14.8K QA pairs** in multiple language varieties, **20K synthetic audio samples** for training, **141  hours of human-recorded audio** for evaluation, and **105K QA pairs manually verified by annotators**.

* **Comprehensive benchmarking** across a diverse set of open-source and proprietary multimodal and audio-based models, offering the community a unified and rigorous evaluation suite.

* **Demonstration of the effectiveness of supervised fine-tuning**, showing consistent performance improvements and confirming the importance of culturally grounded and domain-relevant adaptation for multimodal models.

We sincerely appreciate your time and consideration, and we hope that our clarifications and revisions address all remaining concerns.

Best regards,

The Authors

---

### Meta-Review · Area_Chair_aq3J · 2025-12-26

**Summary:**

EverydayMMQA introduces a novel framework for creating large-scale, culturally grounded datasets and presents OASIS, a tri-modal resource integrating text, images, and speech focused on English and Arabic varieties across 18 countries. The dataset comprises approximately 0.92 million images, 14.8 million QA pairs, and 3.7 million spoken questions, supporting four input modalities (text-only, speech-only, text+image, speech+image) and emphasizing everyday reasoning and cultural nuances. The authors benchmark several closed-source and open-source models, demonstrating that visual grounding significantly enhances performance, particularly for underrepresented languages, and show that fine-tuning on OASIS improves cultural awareness. The framework is designed as language- and location-agnostic, with modules for topic generation, image retrieval, QA generation, and quality checks, incorporating human validation to mitigate biases.

The reviewers raised several critical concerns regarding the paper's methodology and contributions. Key issues include the overreliance on LLMs like GPT-4o for generating and scoring culturally grounded content, which risks perpetuating synthetic or stereotyped data rather than ensuring authenticity. Questions were raised about the generalizability of the framework, with criticisms that it may be a specific pipeline rather than a reusable tool, and fine-tuning experiments were seen as circular without external validation. Data quality was questioned, particularly the use of synthetic speech over real human recordings, which limits real-world applicability, and the non-zero text-only performance in VQA tasks, suggesting that some questions do not adequately require visual grounding. Methodological clarity was lacking, with unclear annotation guidelines and insufficient details on cultural relevance scoring. Additionally, omissions in citing relevant works (e.g., CVQA, PALO) and discrepancies in dataset statistics were noted. These concerns highlight potential weaknesses in the paper's validity, scalability, and cultural authenticity, influencing a cautious assessment of its contributions.

**Reviewer Concerns:**

The authors convincingly demonstrated scalability and utility through external benchmarks and human validation. The addition of 141 hours of human-recorded speech and manual QA checks enhances real-world relevance. The framework's modular design, as illustrated in Figure 2, supports reusability for other languages/regions.

The rebuttal partially alleviates key concerns but leaves some unresolved:

Reliance on LLMs for core tasks (e.g., cultural scoring) remains a latent risk, as human verification may not fully offset model biases. The high variance in query counts per topic (e.g., 10–86) and synthetic speech dominance still limit the dataset's authenticity for noisy, real-world scenarios. While fine-tuning shows gains, the cultural depth of learned representations is unproven.

**Reviewer Scores:**

Reviewer rPig(Initial Score: 4): rPig's primary concerns involved overreliance on GPT-4o for cultural relevance scoring, lack of generalizability testing, and insufficient qualitative error analysis. The authors addressed these by highlighting human-in-the-loop validation, adding external benchmark results (e.g., ALM-Bench), and providing error analyses (e.g., category-wise performance in Appendix figures). Given the thorough rebuttal, rPig might raise his score after discussion, as the responses directly mitigated concerns about LLM bias and demonstrated broader applicability.

Reviewer t9AU (Initial Score: 4): t9AU criticized the framework's novelty (viewing it as a pipeline rather than a reusable tool), automated data generation risks, and circularity in fine-tuning. The authors clarified the modular, language-agnostic design, emphasized human checks, and cited external validation. However, t9AU's confidence in cultural authenticity might remain low due to inherent LLM limitations.  t9AU might maintain the score at 4​ as the rebuttal improved clarity but may not have fully resolved doubts about synthetic data realism.

Reviewer Bqyn (Initial Score: 4): Bqyn focused on methodological clarity, LLM-based cultural scoring validity, and image curation challenges. The authors revised the framework description for better alignment with modules, added dataset examples, and explained human oversight in query filtering. These clarifications likely addressed Bqyn's concerns effectively. Thus, Bqyn might increase his score.

Reviewer Y1YF (Initial Score: 2): Y1YF had strong reservations about the VQA data's visual grounding (non-zero text-only performance), LLM-generated data quality, and query variance. The authors provided annotation guidelines, additional experiments, and arguments aligning with standard VQA practices (e.g., citing Goyal et al.). Y1YF's follow-up comment indicated partial satisfaction but persistent doubts about T+I implications. In a full discussion, Y1YF might have moderated his score to 4, acknowledging improvements but likely maintaining that core issues around language priors were not fully resolved.

---

### Decision · Program_Chairs · 2026-01-26

Reject